# Screening non-conventional yeasts for acid tolerance and engineering *Pichia occidentalis* for production of muconic acid

Michael E. Pyne[1,2,5,10], James A. Bagley [1,6,10], Lauren Narcross[1,2,7], Kaspar Kevvai [1,2,8], Kealan Exley[1,2,9], Meghan Davies[1,2,3], Qingzhao Wang[4], Malcolm Whiteway [1,2] & Vincent J. J. Martin [1,2] ✉

*Saccharomyces cerevisiae* is a workhorse of industrial biotechnology owing to the organism's prominence in alcohol fermentation and the suite of sophisticated genetic tools available to manipulate its metabolism. However, *S. cerevisiae* is not suited to overproduce many bulk bioproducts, as toxicity constrains production at high titers. Here, we employ a high-throughput assay to screen 108 publicly accessible yeast strains for tolerance to $20 \, g \, L^{-1}$ adipic acid (AA), a nylon precursor. We identify 15 tolerant yeasts and select *Pichia occidentalis* for production of *cis,cis*-muconic acid (CCM), the precursor to AA. By developing a genome editing toolkit for *P. occidentalis*, we demonstrate fed-batch production of CCM with a maximum titer ($38.8 \, g \, L^{-1}$), yield ($0.134 \, g \, g^{-1}$ glucose) and productivity ($0.511 \, g \, L^{-1} \, h^{-1}$) that surpasses all metrics achieved using *S. cerevisiae*. This work brings us closer to the industrial bioproduction of AA and underscores the importance of host selection in bioprocessing.

Historically, brewer's yeast (*Saccharomyces cerevisiae*) has been the preferred microbial host for the overproduction of fuels, chemicals, and pharmaceuticals. However, owing to the low value of many bulk chemicals and biofuels, concentrations required for commercial scale production are often toxic to *S. cerevisiae* and other model organisms. For instance, *S. cerevisiae* is unable to sustain growth in low to moderate concentrations of butanol (2%)[1], vanillin (0.05%)[2], benzaldehyde (0.1–0.2%)[3], and many organic acids (0.25–2.5%)[4–6]. Organic acids pose a distinct bioprocessing challenge, as an ideal organic acid bioprocess would be performed at acidic pH, which simplifies downstream recovery of the undissociated form and circumvents the requirement of maintaining pH during fermentation. However, acid toxicity is inversely proportional to pH, as undissociated acids ($pH < pK_a$) freely diffuse into cells and dissociate in the cytosol (pH 5–7)[7]. Accordingly, to circumvent product toxicity, cultivation of most organic-acid-

producing strains of *S. cerevisiae* and other microbes is performed at $pH > pK_a$[8–10].

Adipic acid is a C6 dicarboxylic acid used in the production of nylon 6,6. With an annual production of three million tonnes and a global market of six billion USD[11], adipic acid has been classified as a 'super-commodity' by the U.S. Department of Energy[12]. Global demand for adipic acid is presently met via a chemical process involving cyclohexane and nitric acid, but this method utilizes petrochemical resources and liberates nitrous oxide, a potent greenhouse gas, prompting the investigation of more sustainable routes to its production. There are no natural routes for adipic acid biosynthesis but at least eight engineered biochemical pathways have been proposed for its bioproduction[11]. Of these pathways, major progress has been attained using the *cis,cis*-muconic acid (CCM) route. This strategy employs three heterologous enzymes, 3-dehydroshikimate dehydratase, protocatechuic acid

[1]Department of Biology, Concordia University, Montréal, QC H4B 1R6, Canada. [2]Centre for Applied Synthetic Biology, Concordia University, Montréal, QC H4B 1R6, Canada. [3]BenchSci, Toronto, ON, Canada. [4]bp Biosciences Center, San Diego, California, USA. [5]Present address: Department of Biology, University of Western Ontario, Ontario, Canada. [6]Present address: Centre for Applied Synthetic Biology, Concordia University, Montréal, QC H4B 1R6, Canada. [7]Present address: Amyris, Inc., Emeryville, CA, USA. [8]Present address: Pivot Bio, Berkeley, CA, USA. [9]Present address: Novo Nordisk Foundation Center for Biosustainability, Lyngby, Denmark. [10]These authors contributed equally: Michael E. Pyne, James A. Bagley. ✉e-mail: vincent.martin@concordia.ca

decarboxylase, and catechol dioxygenase, for synthesis of CCM from 3-dehydroshikimate, an intermediate of the shikimate pathway. CCM is converted to adipic acid via hydrogenation by enoate reductase enzymes, yet bacterial variants characterized to date function poorly in yeast hosts[13], achieving <0.9% molar conversion of CCM to adipic acid[14]. In the absence of a suitable yeast-active enoate reductase, *S. cerevisiae* has been engineered to synthesize CCM at a titer, yield, and productivity of up to $22.5\,g\,L^{-1}$, $0.1\,g\,g^{-1}$ glucose, and $0.21\,g\,L^{-1}\,h^{-1}$, respectively[9,10]. Despite these advancements, yeast CCM processes are performed at pH 5−6, which is well above the $pK_a$ of CCM (3.87) and maintains the acid in its dissociated form to limit product toxicity.

In response to the poor tolerance of *S. cerevisiae* to many bulk bioproducts, focus has shifted to screening non-conventional yeast species for phenotypes better suited for industrial applications[15]. Strain selection is a critical and often overlooked aspect of bioprocess development, as selecting a microbial host that is tailored to the target end product is likely to reduce strain engineering interventions, simplify downstream separation, limit product toxicity, and improve overall process metrics[16]. For instance, *Yarrowia lipolytica* is an oleaginous and osmotolerant host that is well-suited for the production of fatty acids and alkanes[17], *Kluyveromyces marxianus* is a rapid-growing thermotolerant yeast[18], and *Pichia kudriavzevii* is a robust acid tolerant host utilized for the production of diverse organic acids[19]. Approximately 1500 species of yeast have been classified across more than 100 genera and public culture repositories contain thousands of distinct yeast strains. The majority of these yeast species can be cultivated in the laboratory with standard media[20] and major advancements in genome sequencing, systems biology, and strain engineering have dramatically reduced the cost and labor involved in the genetic domestication of non-conventional hosts. Sophisticated genetic toolkits have been developed for diverse yeast species, including *Y. lipolytica*[21], *K. marxianus*[18], and *P. pastoris*[22], providing new opportunities for host selection beyond *Saccharomyces*.

In this study, we screen 108 distinct strains of yeast from public culture collections for natural tolerance to high concentrations of adipic acid and other dicarboxylic acids. Our screen identifies several species of *Pichia* as prospective industrial hosts to produce diverse organic acids. Based on antibiotic susceptibility, plasmid transformation, and CRISPR-Cas9 editing assays, we select *P. occidentalis* for high level production of CCM, the direct precursor to adipic acid. Under fed-batch conditions, engineered *P. occidentalis* synthesizes $38.8\,g\,L^{-1}$ CCM with a maximum yield and productivity that surpasses all present metrics achieved using *S. cerevisiae*.

## Results

### Screening non-conventional yeast strains for organic acid tolerance

To build a library of non-conventional yeasts for screening prospective acid tolerant hosts, we surveyed public culture repositories for non-*Saccharomyces* strains, selecting strains with reported acid tolerance where available. In total, 153 strains from 83 distinct species were obtained, of which 124 were successfully cultured in YM medium at 30 °C and 108 were successfully screened in our assay, with the remainder being unable to grow reliably in YPD. The final 108 strains spanned 66 species from 28 genera (Supplementary Data 1).

Subsequent screening involved cultivating strains in YPD containing $20\,g\,L^{-1}$ citric acid (0.1 M; $YPD_{cit}$; Fig. 1a) or YPD containing $20\,g\,L^{-1}$ adipic acid (0.137 M; $YPD_{AA}$; Fig. 1b). Addition of adipic acid dropped the pH of YPD to ~3.7 ($pK_{a1} = 4.41$, $pK_{a2} = 5.41$) and addition of citric acid dropped the pH of YPD to ~3.0 ($pK_{a1} = 3.13$, $pK_{a2} = 4.76$, $pK_{a3} = 6.39$). $YPD_{AA}$ medium yielded a wide diversity of growth phenotypes. Despite a higher pH, the correlation between growth in YPD and $YPD_{AA}$ ($R^2 = 0.685$) was lower than that of growth in YPD and $YPD_{cit}$ ($R^2 = 0.905$), indicating that adipic acid impacted the growth of more strains.

We employed two criteria for selecting strains for subsequent rounds of screening. First, we defined adipic acid tolerance as a growth rate in $YPD_{AA}$ (estimated as area under the curve at $OD_{600}$) that is no more than 20% slower compared to growth in YPD. Second, to ensure robust growth of strains under standard laboratory conditions, we selected strains with growth rates in YPD (measured as area under strain growth curves at $OD_{600}$) that were no more than 20% slower compared to the control strain (*S. cerevisiae* CEN.PK 113-7D). Despite the use of rich YPD medium, some strains were unable to grow to an $OD_{600}$ >1.0 in the allotted time in the absence of acid inhibitors.

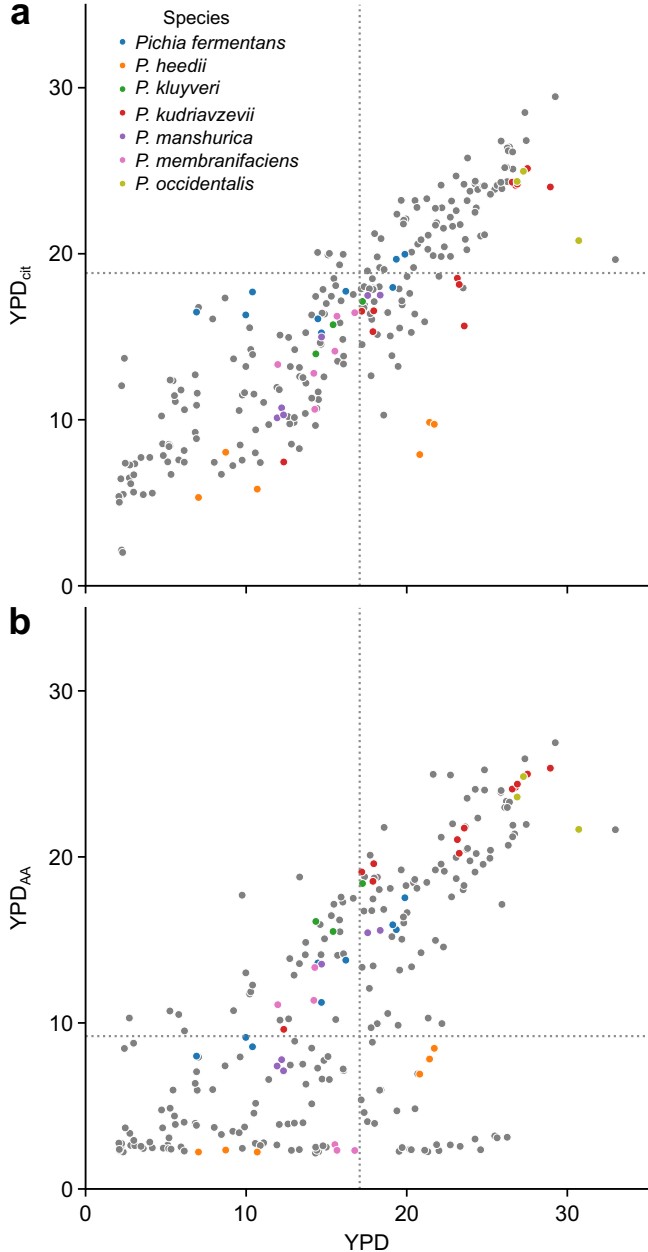

**Fig. 1 | Growth of 83 candidate yeast strains in YPD supplemented with citric and adipic acids. a** Growth of yeast strains in YPD compared to YPD supplemented with 0.1 M citric acid ($YPD_{cit}$). **b** Growth of yeast strains in YPD supplemented with 0.15 M adipic acid ($YPD_{AA}$). Growth was assessed by measuring area under the curve (AUC) of growth curves. Each data point represents a single replicate of *n* = 3 independent biological replicates for each yeast strain. Dashed lines show average AUCs for *S. cerevisiae* CEN.PK113-7D in each growth condition. Source data are provided as a Source Data file.

Applying our screening criteria resulted in the identification of 15 fast growing and adipic acid tolerant strains from the original set of 108 candidates (Table 1). Notably, strains from the genus *Pichia* were enriched among the adipic acid tolerant candidates; of the fifteen strains, seven were *Pichia* (47%), while *Pichia* represented only 10% of the original strain collection.

### Table 1 | Summary of 15 adipic acid tolerant yeast identified from a preliminary screen

| Strain ID | Depositor ID | Species | YPD AUC[a] | YPD_AA AUC[a] |
|---|---|---|---|---|
| 00E | 3092 CBS | *Candida boidinii* | 21.6 | 19.7 |
| 019 | Y-7921 NRRL | *Candida sorbophila* | 28.5 | 24.7 |
| 01L | 8880 CBS | *Cyberlindnera saturnus* | 18.7 | 18.7 |
| 01R | 2530 NCYC | *Debaryomyces hansenii* | 19.9 | 20.9 |
| 01X | 1478 NCYC | *Kazachstania exigua* | 25.6 | 21.4 |
| 020 | 971 NCYC | *K. unispora* | 17.8 | 17.4 |
| 02G | 562 NCYC | *Pichia fermentans* | 20.4 | 16.6 |
| 01W | 4001 NCYC | *P. kudriavzevii* | 30.5 | 26.5 |
| 02M | 2658 NCYC | *P. kudriavzevii* | 26.6 | 22.2 |
| 01V | 55 NCYC | *P. kudriavzevii* | 19.5 | 20.5 |
| 02N | 872 NCYC | *P. kudriavzevii* | 18.2 | 19.9 |
| 02Q | Y-27978 NRRL | *P. manshurica* | 17.3 | 14.8 |
| 02 W | Y-7552 | *P. occidentalis* | 29.7 | 25.0 |
| 03 P | Y-27308 NRRL | *Tetrapisispora arboricola* | 22.6 | 19.5 |
| 03 V | 432 NCYC | *Wickerhamomyces anomalus* | 27.1 | 26.1 |

[a]Relative growth rates were compared by measuring area under the curve (AUC).

The over-representation of *Pichia* strains prompted further probing of the *Pichia* clade for acid tolerant hosts. An additional eight *Pichia* strains were added, three of which were from species not represented in the initial screen (*Candida sorboxylosa*, *Pichia norvegensis* and *Pichia exigua*). Despite its name, *C. sorboxylosa* has been placed in the *Pichia* genus by whole genome phylogenetic analysis[23]. We also added two new strains of *P. occidentalis* (Y-6545 and YB-3389) based on the adipic acid tolerance of *P. occidentalis* Y-7552 in our initial screen.

Subsequent acid tolerance screening was performed in defined Yeast Nitrogen Base (YNB) medium to identify strains with robust growth in both defined and rich media. Intriguingly, certain *P. occidentalis* strains displayed a growth defect in YNB. Specifically, *P. occidentalis* Y-6545 and YB-3389 were unable to reach an $OD_{600}$ of 1.0 in the allotted time (24 h) in 1× YNB compared to an $OD_{600}$ of ~1.75 in YPD (Supplementary Fig. 1). An increase in the concentration of YNB from 1× (1.7 g L$^{-1}$) to 3× (5.1 g L$^{-1}$) alleviated this limitation. Since YNB is composed of a mixture of 20 vitamins and salts, the nature of this apparent nutrient limitation is not known. To ensure robust growth of all yeast strains, subsequent screening experiments were performed in 3× YNB.

To further characterize acid tolerant phenotypes, we performed a subsequent round of screening in defined medium to evaluate the ability of 10 candidate *Pichia* strains to grow on three different dicarboxylic acids of varying chain lengths, namely succinic (C4), glutaric (C5), and adipic acid (C6) (Fig. 2a). Acids were added to each defined medium at a concentration of 0.15 M and a pH of 2.8. For the dicarboxylic acids tested, the control *S. cerevisiae* CEN.PK 113-7D strain exhibited a steady decrease in growth rate with increasing acid chain length assuming a comparable level of dissociation between the dicarboxylic acids. Adipic acid, the longest dicarboxylic acid screened,

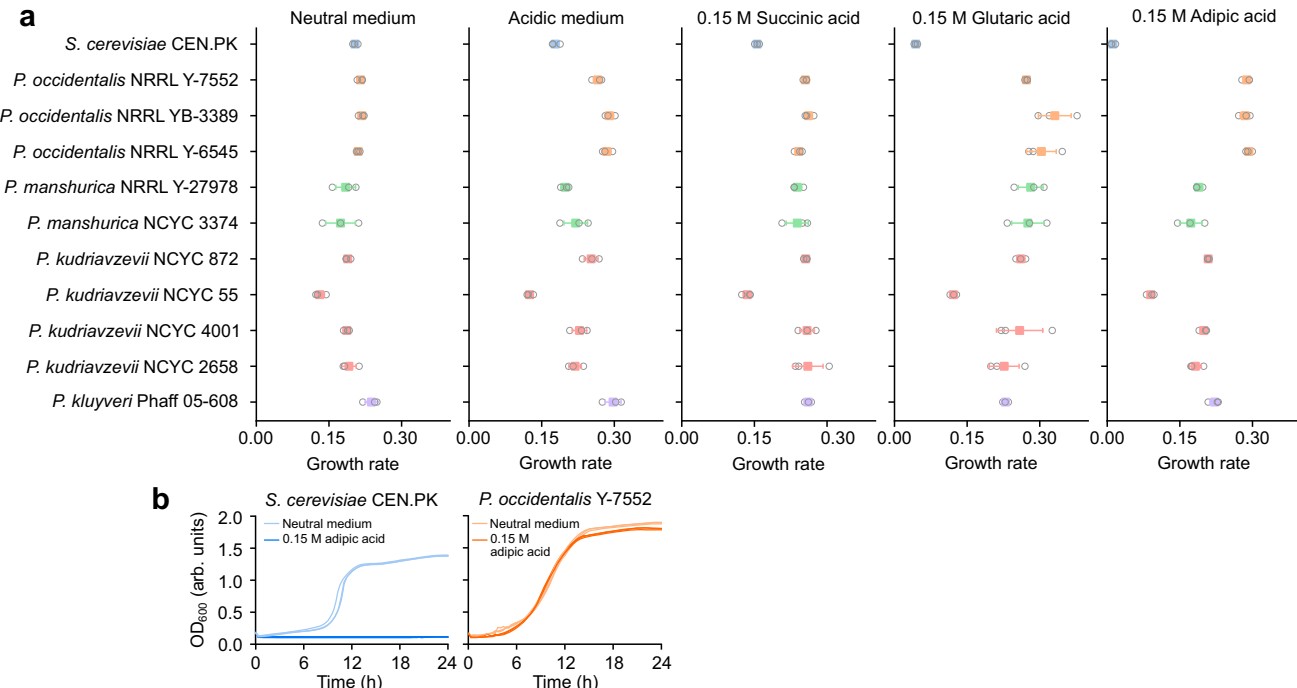

**Fig. 2 | Growth of 10 candidate *Pichia* strains in various dicarboxylic acids.** **a** Maximum growth rates of candidate strains in 3× YNB supplemented with succinic, glutaric, or adipic acids (0.15 M each). All dicarboxylic-acid-containing media were pH adjusted to 2.8 after the addition of 0.15 M of the respective dicarboxylic acid. Acidic medium is 3× YNB medium with the pH adjusted to 2.8 and neutral medium is 3× YNB medium without pH adjustment. Error bars represent the mean ± s.d. of $n = 3$ independent biological samples. **b** Representative growth curves of *S. cerevisiae* CEN.PK and *P. occidentalis* strain Y-7552 in 3× YNB supplemented with 0.15 M (22 g L$^{-1}$) adipic acid. Growth was assessed by measuring $OD_{600}$ in arbitrary units (arb. units). Three independent biological replicates are overlaid for each growth condition. Source data are provided as a Source Data file.

completely inhibited growth of *S. cerevisiae* at a concentration of 0.15 M (21.9 g L$^{-1}$) (Fig. 2b). This general trend was also observed with most of the *Pichia* strains, except for the three strains of *P. occidentalis*, where the growth rate generally increased with chain length (*P* < 0.001) (Fig. 2a). Overall, one or multiple strains of *P. kluyveri*, *P. kudriavzevii*, *P. manshurica*, and *P. occidentalis* exhibited notable tolerance to multiple dicarboxylic acids.

We also screened 93 yeast strains in YPD medium and YPD containing 0.1 M citrate buffer, pH 3.0 (YPD$_{cit}$). However, this condition was not sufficiently inhibitory, as most strains grew to comparable levels in YPD and YPD$_{cit}$ (Fig. 1a). It has been reported that citric acid is more toxic to yeast at pH 4.5 than pH 3.0 owing to the propensity of the dissociated form to bind divalent cations, providing a possible explanation for our observations.

In addition to dicarboxylic acids, we screened 26 *Pichia* strains on several monocarboxylic acids using the same conditions (defined 3× YNB medium containing 0.15 M of acid, pH 2.8). Only one strain grew in acetic acid (*P. manshurica* Y-27978) and none of the tested strains grew in propionic, butyric or valeric acids. Assuming comparable levels of dissociation between monocarboxylic acids, these findings indicate a higher degree of toxicity caused by propionic, butyric and valeric acids at equimolar concentrations.

### Additional traits used for selection

Based on our screen for tolerance to adipic acid, we selected eight candidate strains of *Pichia* as potential CCM production hosts. We first assessed carbon source utilization of the eight selected strains by testing the capacity to utilize xylose, an abundant C5 sugar in plant lignocellulose. Screening xylose utilization by patching colonies onto agar plates containing xylose as sole carbon source revealed that *P. fermentans* 562 NCYC was the only adipic acid tolerant *Pichia* strain capable of xylose utilization (Fig. 3a), which is consistent with yeast phenotypic characterization data[20] (Supplementary Table 1). In addition to carbon source utilization, strain suitability for genetic and metabolic engineering requires the ability to use common selectable markers and introduce foreign genetic material. Accordingly, the eight selected *Pichia* strains were screened for susceptibility to high concentrations of G418 (1000 µg mL$^{-1}$), hygromycin (600 µg mL$^{-1}$), nourseothricin (100 µg mL$^{-1}$), and zeocin (300 µg mL$^{-1}$) (Fig. 3a). Except for *P. membranifaciens* NCYC 55 and *P. manshurica* Y-27978, all strains were resistant to 300 µg mL$^{-1}$ zeocin. In contrast, nearly all strains were susceptible to nourseothricin and hygromycin at the concentrations tested. G418 was effective against roughly half of the strains assayed. Of the three strains of *P. occidentalis*, only strain Y-7552 was susceptible to G418, while nourseothricin and hygromycin were effective against all three strains.

Having identified effective concentrations of antibiotics, we screened the eight candidate *Pichia* strains for transformation using vectors expressing a hygromycin-resistance marker from the *Ashbya gossypii TEF1* promoter (P$_{TEF1}$). We confirmed successful transformation of six of the eight strains of *Pichia* (Fig. 3a). The highest transformation efficiency was attained using *P. fermentans*, the least adipic acid tolerant of the eight candidate *Pichia* strains. Transformants were obtained for all three strains of *P. occidentalis*, the most adipic acid tolerant strains identified in this study. Transformation efficiency of *P. occidentalis* Y-7552 (10$^2$ CFU µg$^{-1}$ DNA) was higher than strains Y-6545 and YB-3389 (10$^1$ CFU µg$^{-1}$ DNA).

Based on antibiotic susceptibility and plasmid transformation assays, we selected *P. kluyveri* and all three strains of *P. occidentalis* for genome editing assays, which was aided by recently reported draft genome sequences for these strains[24]. We selected the *ADE2* gene for deletion since its inactivation leads to accumulation of a colored intermediate for visual identification of deletion mutants (Fig. 3b). We anticipated a low level of homologous recombination in *Pichia* strains[25,26], prompting us to pursue a CRISPR system in which

homology-directed repair templates were precloned into pCas delivery vectors (Fig. 3c), thus circumventing the need to assemble multiple overlapping DNA fragments in *Pichia*.

We first assessed the capacity of an in-house CRISPR-Cas9 vector (pCas-Hyg-CEN6ARS4) possessing the *S. cerevisiae* CEN6/ARS4 origin, as well as gRNA and *cas9* expression cassettes (Fig. 3c), to function in *Pichia*. We designed *ade2Δ* homology repair templates for all four candidate *Pichia* strains by fusing ~800 bp homology arms flanking the *ADE2* coding sequence and small portions (~100–150 bp) of the promoter and terminator elements (Fig. 3b). Using this design, homology-directed repair of an *ADE2* double-strand break results in a large deletion (~2 kb) of the entire *ADE2* coding sequence. Following assembly of *ade2Δ* pCas vectors in *S. cerevisiae*, vectors were linearized with BglII and used to transform the relevant *Pichia* strain along with an overlapping *ADE2* gRNA for assembly via gap repair in *Pichia* (Fig. 3d). Using this approach, we obtained Hyg$^R$ colonies for all four strains of *Pichia* (Fig. 3e), signifying successful vector assembly via gap repair. Following screening of the *ADE2* locus, we identified *ade2Δ* colonies in all three strains of *P. occidentalis*, but we were unable to identify *ade2Δ* mutants from Hyg$^R$ colonies of *P. kluyveri* (Fig. 3f). Editing efficiencies varied from 50% to 69% based on screening of 10–13 Hyg$^R$ transformants selected at random.

Taken together, *P. occidentalis* Y-7552 was selected as the final host for production of CCM. Strain Y-7552 is sensitive to at least three antibiotics (G418, hygromycin, and nourseothricin), yields the highest transformation efficiency of the three *P. occidentalis* strains (Fig. 3a), and does not exhibit the growth defect in 1× YNB observed for strains Y-6545 and YB-3389 (Supplementary Fig. 1). Strain Y-7552 is diploid and the type strain of *P. occidentalis*[27]. The strain is able to utilize glucose, glycerol, and ethanol as sole carbon sources, yet does not grow on sucrose, galactose, lactose, xylose, starch, or cellobiose[20] (Supplementary Table 1). In addition to *P. kudriavzevii*, *P. occidentalis* grows in medium devoid of common yeast vitamins, adding to the strain's potential for industrial production of high value organic acids.

### CRISPR-Cas9 editing and antibiotic marker recycling in *P. occidentalis* Y-7552

Having successfully deleted the *ADE2* gene in *P. occidentalis* strain Y-7552 using CRISPR-Cas9, we wished to cure cells of the pCas-Hyg-ADE2 gRNA vector, which is a requisite step for performing iterative rounds of CRISPR-Cas9 genome editing. Because the CEN6/ARS4 replication module is derived from *Saccharomyces cerevisiae*, we anticipated chromosomal integration of the pCas-Hyg-CEN6ARS4 vector in *Pichia*. Accordingly, the vector could not be cured by subculturing up to six times in non-selective YPD medium, and other methods of plasmid curing were unsuccessful, including culturing at an elevated temperature (37 °C), heat-shocking cells at 42 °C in a mock lithium-acetate-PEG transformation, or attempting to displace the original Hyg$^R$-CEN6-ARS4 vector by introducing a G418$^R$-CEN6-ARS4 or Nat$^R$-CEN6-ARS4 vector to the *ade2Δ* host (Supplementary Fig. 2). We also constructed pCas-G418-ADE2 vectors containing the *S. cerevisiae* 2µ origin (pCas-G418-2µ-ADE2) and the broad host panARS origin (pCas-G418-panARS-ADE2), and while both vectors facilitated deletion of *ADE2* based on color phenotype and colony PCR, all transformants exhibited stable integration of the selectable markers (Supplementary Fig. 3).

To confirm vector integration, we constructed a pCas derivative lacking a functional plasmid origin (pCas-G418-NoOri-ADE2). Introduction of pCas-G418-NoOri-ADE2 to *P. occidentalis* generated G418-resistant transformants, including red *ade2Δ* colonies (Supplementary Fig. 4), indicating chromosomal integration as the vector lacks a functional replication origin. Collectively, these data are consistent with chromosomal integration of pCas vectors in *P. occidentalis*.

Chromosomal integration of vectors poses a challenge for strain engineering, as chromosomal markers are difficult to excise from cells,

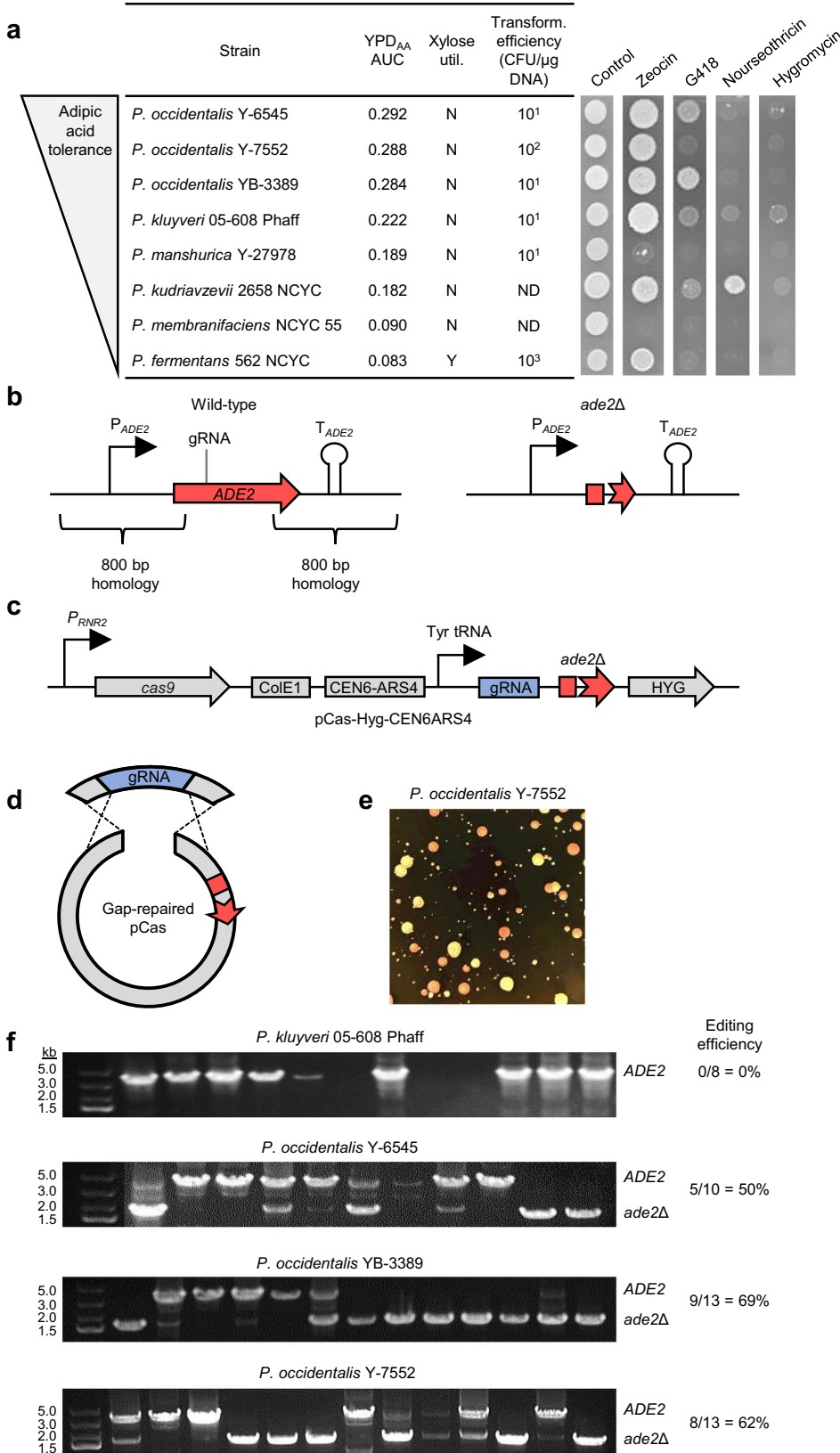

thus hindering subsequent rounds of genome editing. To enable the recycling of antibiotic-resistance markers in *P. occidentalis*, we explored CRISPR-Cas9 as a means of inactivating or swapping chromosomal antibiotic resistance markers. We first deleted the *ADE2* gene using pCas-G418-CEN6ARS4-ADE2, yielding an *ade2*Δ G418[R] mutant. In a subsequent editing event, we introduced to the *ade2*Δ G418[R] strain a

pCas-Hyg-CEN6ARS4 vector harboring a gRNA targeting the G418-resistance marker. Targeting the chromosomal G418[R] marker yielded a substantial increase in the number of Hyg[R] transformants compared to a control transformation in which the G418 gRNA was replaced with a non-targeting gRNA (Supplementary Fig. 5). While we did not include an exogenous donor for repair of double-stranded breaks (DSBs)

**Fig. 3 | Xylose utilization, antibiotic susceptibility, plasmid transformation, and CRISPR-Cas9 editing of adipic acid tolerant *Pichia*. a** Xylose utilization, antibiotic susceptibility and transformation efficiency of adipic acid tolerant *Pichia* spp. Strains are sorted by adipic acid tolerance (AUC in YPD containing 20 g L$^{-1}$ adipic acid) and xylose utilization and susceptibility to four common yeast anti-biotics is shown. Transformation efficiency using a plasmid conferring Hyg$^R$ is provided rounded to the nearest order of magnitude. Transformants of *P. kudriavzevii* 2658 NCYC and *P. membranifaciens* NCYC55 were not detected (ND). Repeating plasmid transformation and antibiotic susceptibility assays routinely yielded similar results. **b** *ADE2* deletion strategy utilized in adipic acid tolerant *Pichia*. Large homology regions (~800 bp) were designed flanking the *ADE2* coding sequence. **c** Structure of *ADE2* pCas CRISPR-Cas9 vectors containing precloned *ade2*Δ donor cassettes. Donor cassettes were precloned into the BglII site of pCas vectors. **d** Gap repair strategy of pCas vector assembly in adipic acid tolerant *Pichia*. pCas vectors harboring precloned *ade2*Δ donor cassettes were linearized by digestion with BsaI or NotI and used to transform *Pichia* species along with an overlapping *ADE2* gRNA PCR product. **e** Deletion of *ADE2* yields pink colonies in adipic acid tolerant *Pichia*. Transformants were plated onto YPD agar plates containing hygromycin without adenine supplementation. A representative YPD agar plate containing *P. occidentalis* Y-7552 is shown. **f** Screening *ADE2* deletion in adipic acid tolerant *Pichia*. A pCas-Hyg-CEN6ARS4 vector possessing an *ADE2* gRNA and repair donor was transferred to *P. kluyveri* and three strains of *P. occidentalis*. The *ADE2* locus was screened for deletion in randomly selected Hyg$^R$ transformants prior to color development. Source data are provided as a Source Data file.

delivered to the chromosomal G418 marker, we suspected that the newly introduced pCas-Hyg-CEN6ARS4 vector provided an efficient DSB repair donor. Specifically, pCas-Hyg-CEN6ARS4 harbors a vector-encoded P$_{TEF1}$-Hyg-T$_{TEF1}$ donor for repair of the DSB delivered to the chromosomal P$_{TEF1}$-G418-T$_{TEF1}$ cassette within the *ade2*Δ G418$^R$ strain. Loss of G418 resistance was confirmed by absence of growth on G418-containing YPD agar plates and by PCR using primers specific to the Hyg$^R$ marker (Supplementary Fig. 5).

Having established a general strategy for recycling antibiotic markers, we tested this approach for performing iterative rounds of CRISPR editing in *P. occidentalis* by using our *ade2*Δ G418$^R$ mutant to delete a new target gene and concurrently inactivate the pre-existing G418$^R$ marker. For gene deletion, we selected *FCY1*, which encodes cytosine deaminase, as its inactivation confers resistance to 5-fluor-ocytosine, a toxic analog of cytosine (Supplementary Fig. 6). Deleting *FCY1* and eliminating G418$^R$ requires the introduction of two gRNAs from a single pCas-Hyg-CEN6ARS4 vector. To test the delivery and efficiency of two gRNA species, we transformed the *ade2*Δ G418$^R$ mutant with linearized pCas-Hyg-CEN6ARS4 vector along with *FCY1* and G418 gRNAs, each possessing overlap to pCas-Hyg-CEN6ARS4. Because only one gRNA species is required for gap repair, Hyg$^R$ colonies possess three possible genotypes: (1) *FCY1* G418$^S$ single mutants; (2) *fcy1*Δ G418$^R$ single mutants; or (3) *fcy1*Δ G418$^S$ double mutants. To assay the proportion of these genotypes, we introduced equal amounts of *FCY1* and G418 gRNAs and screened resulting Hyg$^R$ colonies for *FCY1* deletion and G418 susceptibility. Screening 18 Hyg$^R$ colonies in this manner yielded 11 G418$^S$ colonies, but no *fcy1*Δ mutants, indicating that gap repair occurred exclusively using the G418 gRNA despite the presence of equal amounts of G418 and *FCY1* gRNAs (Supplementary Fig. 7).

Because targeting a chromosomal G418$^R$ marker yields a major increase in the number of Hyg$^R$ transformants (Supplementary Fig. 5), we surmised that the transformation advantage afforded by the G418 gRNA obscured *fcy1*Δ mutant colonies, yielding exclusively *FCY1* G418$^S$ single mutants. To overcome this imbalance, we titrated the relative amount of G418 and *FCY1* gRNA species. We transformed the *ade2*Δ G418$^R$ mutant with a 20-, 10-, 5-, and 3-fold excess of *FCY1* gRNA relative to G418 gRNA to offset the advantage in transformation afforded by the G418 gRNA. Screening resultant Hyg$^R$ colonies for *FCY1* deletion and G418 susceptibility yielded several colonies possessing both *fcy1*Δ G418$^S$ editing events (Supplementary Fig. 8). The greatest proportion of *fcy1*Δ G418$^S$ colonies (37.5%) was attained by introducing a 10-fold excess of *FCY1* gRNA, while a five-fold and three-fold excess each yielded 27% and 25% double-edited colonies, respectively. A 20-fold excess of *FCY1* gRNA favored *FCY1* deletion, as only 12.5% of colonies possessed both *fcy1*Δ G418$^S$ editing events.

## Characterization of *P. occidentalis* promoters and terminators for strain engineering

To develop further tools for engineering *P. occidentalis* for production of heterologous metabolites, we characterized a panel of native pro-moter and terminator elements. We selected a suitable fluorescent reporter by chromosomally integrating a set of four green fluorescent protein (GFP) gene variants using the *P. occidentalis TDH3* promoter (P$_{TDH3}$), the ortholog of the strongest native promoter in *S. cerevisiae*[28,29]. Enhanced GFP (eGFP), superfolder GFP (sfGFP), Envy GFP, and mNeonGreen GFP were integrated into the *FCY1* locus of *P. occidentalis* using pCas-Hyg-CENARS4-FCY1. Cultures harboring mNeonGreen generated the greatest fluorescence followed by Envy, sfGFP, and eGFP (Fig. 4a).

We then selected a panel of 12 *P. occidentalis* promoters and ter-minators based on those characterized in *S. cerevisiae*[28–30] (Supple-mentary Data 2). Promoter constructs were cloned upstream of mNeonGreen with a fixed *RPL3* terminator, while terminators were cloned downstream of mNeonGreen with a fixed *GPM1* promoter. Full mNeonGreen expression cassettes were cloned into vector pCas-Hyg-CEN6ARS4 flanked by homology to the *P. occidentalis FCY1* gene and integrated into the *FCY1* locus. The fluorescence output from mNeonGreen promoter and terminator integrants was quantified from 12-h exponential-phase cultures. Fluorescence output varied by a fac-tor of 7 and 6.5 across the 12 selected promoters and terminators, respectively (Fig. 4b, c), and by a factor of nearly 20 across all 24 constructs, demonstrating effective modulation of gene expression in *P. occidentalis*. In line with *S. cerevisiae* and other organisms, expres-sion of mNeonGreen from P$_{TDH3}$ yielded the highest fluorescence. T$_{RPL3}$ and T$_{VMA2}$ facilitated high-level expression in *P. occidentalis* and were among the top-performing elements of all 5302 terminators assayed in *S. cerevisiae*[30].

## Engineering *P. occidentalis* for production of *cis,cis*-muco-nic acid

The techniques for CRISPR-Cas9 genome editing, antibiotic marker recycling, and heterologous gene expression in *P. occidentalis*, allowed us to engineer the host for heterologous production of toxic acid products. Although *P. occidentalis* was selected for its tolerance to adipic acid, yeast adipic acid synthesis is currently poor (2.59 mg L$^{-1}$)[14] owing to an inability to functionally express bacterial enoate reductase enzymes in eukaryotic hosts[13,14]. In the absence of such a yeast-active enoate reductase enzyme, we opted to engineer *P. occidentalis* for synthesis of CCM, the direct precursor to adipic acid.

We first engineered *P. occidentalis* Y-7552 to produce proto-catechuic acid (PCA), the committed heterologous metabolite in the proposed CCM synthesis pathway (Fig. 5a). We integrated *AROZ* encoding DHS dehydratase from *Podospora anserina* (Pa*AROZ*) into the *ARO4* locus of *P. occidentalis*. The resulting strain (LP620) pro-duced 1.0 g L$^{-1}$ PCA in batch microtiter plate cultivations from 20 g L$^{-1}$ glucose (Fig. 5b). We next integrated a feedback-resistant mutant of *P. occidentalis* 3-deoxy-D-arabino-heptulosonate-7-phosphate (DAHP) synthase (*ARO4^{K229L}*)[31] into LP620 (yielding LP622) which increased the PCA titer to 1.9 g L$^{-1}$. Subsequent integration of *aroB* and *aroD* from *E. coli*, encoding 3-dehydroquinate synthase and 3-dehydroquinate dehydratase, respectively, into the *P. occidentalis ARO3* locus of strain LP622 (yielding LP630) facilitated a further increase in PCA titer to 2.0 g L$^{-1}$ in batch microtiter plate cultivations (Fig. 5b).

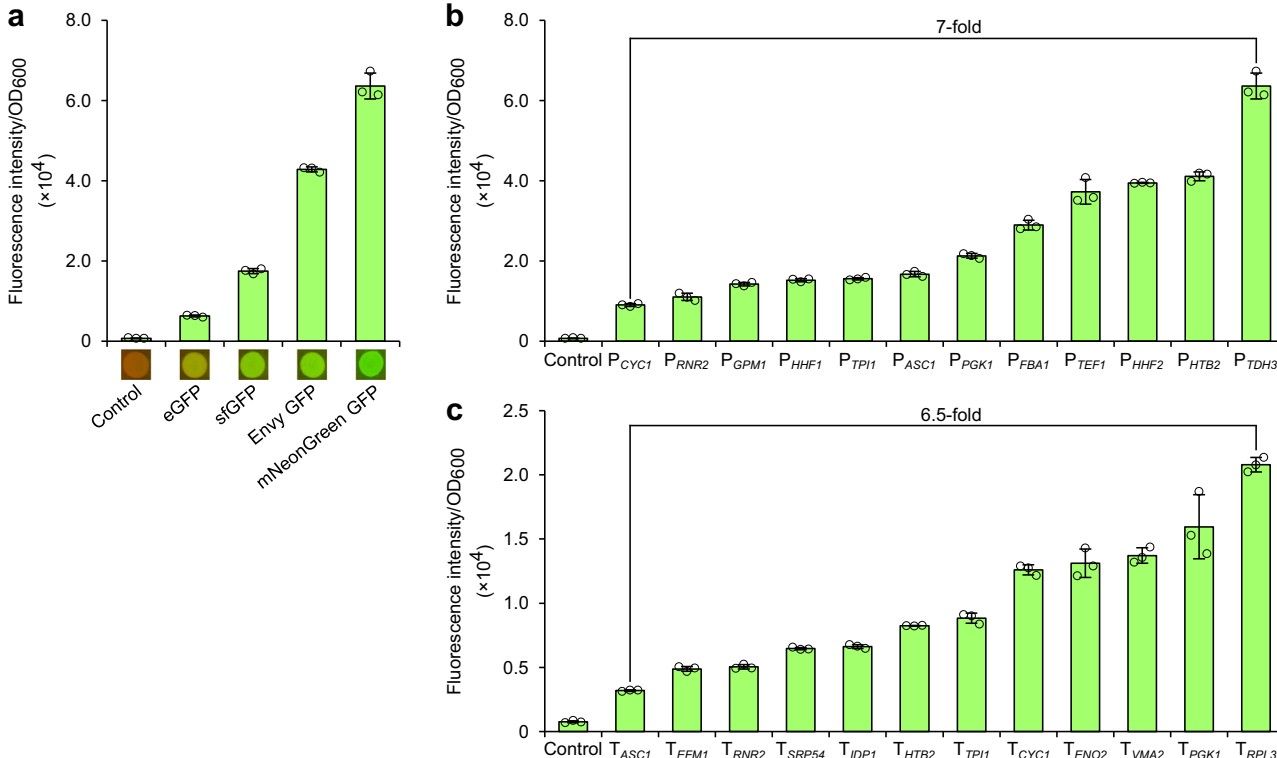

**Fig. 4 | Characterization of *P. occidentalis* promoters and terminators. a** OD-normalized fluorescence intensity of *P. occidentalis* Y-7552 strains expressing common *GFP* gene variants. GFP-encoding variants were expressed using $P_{TDH3}$ and $T_{RPL3}$ from *P. occidentalis* Y-7552. *GFP*-expressing colonies are shown with UV exposure. **b** OD-normalized fluorescence intensity of mNeonGreen expressed from 12 *P. occidentalis* gene promoters with the *RPL3* terminator. **c** OD-normalized fluorescence intensity of mNeonGreen expressed from the *GPM1* promoter and 12 *P. occidentalis* gene terminators. Error bars represent the mean ± s.d. of *n* = 3 independent biological samples. All GFP constructs were integrated into the *FCY1* locus of *P. occidentalis*. Source data are provided as a Source Data file.

In the heterologous pathway, PCA decarboxylase (AroY) converts PCA to catechol, which is subsequently converted to CCM by 1,2-catechol dioxygenase (HQD2). We screened two bacterial AroY variants (AroY from *Klebsiella pneumoniae* and EcdC from *Enterobacter cloacae*) for conversion of PCA to catechol in strain LP630. PCA decarboxylases require a prenylated FMN cofactor that is synthesized by a flavin prenyltransferase in *S. cerevisiae* (*Sc*Pad1)[32]. We were unable to identify a candidate *PAD1*-encoding gene in the genome of *P. occidentalis*, prompting us to investigate pairing of *aroY* and *ecdC* with *ScPAD1*. Analysis of PCA decarboxylase strains revealed production of 1.7 g L$^{-1}$ (*aroY* + *ScPAD1*) and 1.3 g L$^{-1}$ (*ecdC* + *ScPAD1*) catechol and near-complete consumption of PCA (Fig. 5b; Supplementary Fig. 9). By contrast, omission of *ScPAD1* failed to alter PCA levels relative to the control strain (Supplementary Fig. 9), and pairing *aroY* or *ecdC* with *HQD2* from *Candida albicans* (*CaHQD2*) in the absence of *ScPAD1* did not yield catechol or CCM.

To complete the heterologous CCM production pathway, we introduced *CaHQD2* to our catechol production strain expressing *aroY* + *ScPAD1* (LP632). Because *S. cerevisiae* CCM production strains accumulate PCA under fed-batch conditions[9,10], we included an additional copy of *aroY* with *CaHQD2*, yielding strain LP635. The introduction of *aroY* + *CaHQD2* resulted in the production of 2.5 g L$^{-1}$ CCM with a minor quantity of PCA (0.06 g L$^{-1}$) and no detectable catechol (Fig. 5b).

### Fed-batch production of *cis,cis*-muconic acid

After establishing CCM synthesis in 96-well plate format, we characterized our *P. occidentalis* CCM production strain (LP635) in fed-batch fermentation. We first assessed CCM production under cultivation at pH 6.0 to benchmark our *P. occidentalis* host against prior studies involving analogous engineered strains of *S. cerevisiae*[9,10].

Under these conditions, strain LP635 synthesized 38.8 g L$^{-1}$ CCM with a yield and productivity of 0.134 g g$^{-1}$ glucose and 0.511 g L$^{-1}$ h$^{-1}$, respectively (Fig. 5c). A small amount of PCA and catechol was detected (<0.4 g L$^{-1}$ combined concentration).

Having established high-level production of CCM at intermediate pH, we next characterized our CCM production strain without pH control. Under these conditions, the pH rapidly dropped to 2.0 during the batch phase and growth stalled, as evidenced by a decline in $CO_2$ production (Supplementary Fig. 10). Growth could be transiently restored in a pH-dependent manner through stepwise increase in pH from 2.0 to 4.5 in 0.5-unit increments. For instance, increasing culture pH from 3.0 to 3.5 restored growth, as evidenced by a near instantaneous restoration of $CO_2$ production. However, growth was transient, as $CO_2$ concentration began to decline approximately 6 h following the increase in pH. Growth could again be restored by increasing pH from 3.5 to 4.0. Under these conditions, strain LP635 synthesized 7.2 g L$^{-1}$ CCM. Glycerol accumulated throughout the cultivation, reaching a concentration of 12.4 g L$^{-1}$ in late-stage samples.

Although *P. occidentalis* was selected for its robust tolerance to adipic acid at low pH, we surmised that poor growth of our CCM production strain at low pH was a consequence of CCM toxicity. To test this hypothesis, we first compared growth curves of wild-type *P. occidentalis* and strains engineered for PCA and CCM production in 3× YNB (Supplementary Fig. 11). Our engineered PCA-producing strain exhibited a decline in maximum growth rate compared to wild-type *P. occidentalis* ($\mu_{max}$ 0.23 h$^{-1}$ and 0.32 h$^{-1}$, respectively), yet final biomass concentration was largely unaltered between the two strains ($OD_{max}$ 1.8 and 1.9, respectively). The same engineered strain also exhibited a prolonged lag phase relative to the wild-type strain. In contrast, engineering *P. occidentalis* for CCM production caused a 62% and 32% reduction in maximal growth rate and biomass concentration,

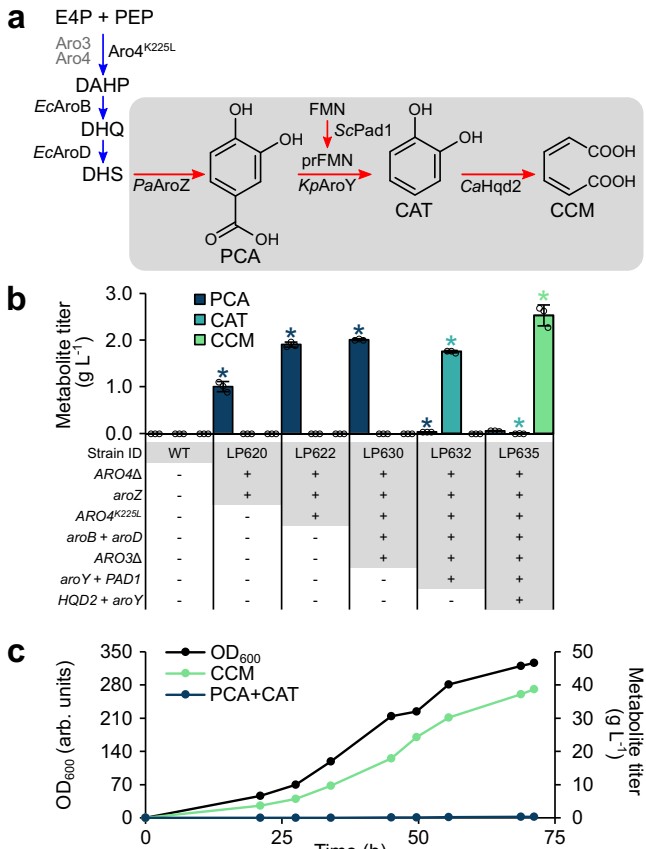

**Fig. 5 | Engineering *P. occidentalis* Y-7552 for efficient synthesis of CCM.**
**a** Heterologous pathway for synthesis of CCM from endogenous
3-dehydroshikimate (DHS). The native yeast shikimate pathway is shown in blue,
while the heterologous CCM synthesis pathway is shaded and shown in red. **b** Titers
of CCM pathway intermediates in culture supernatants of successive engineered *P. occidentalis* production strains. Metabolite titers are depicted in g L$^{-1}$ for PCA, CAT,
and CCM. Asterisks (*) denote a significant increase or decrease in metabolite titer
relative to the respective parent strain ($P < 0.05$). Statistical differences between
parent and derivative strains were tested using two-tailed Student's *t*-test. Error
bars represent the mean ± s.d. of $n = 3$ independent biological samples.
**c** Cultivation of a CCM-producing *P. occidentalis* strain (LP635) in a fed-batch fermentor using a mineral medium at pH 6.0. Growth of biomass (OD$_{600}$) and accumulation of CCM pathway metabolites in the culture medium were measured
during cultivation. OD$_{600}$ is reported in arbitrary units (arb. units). CAT catechol,
CCM *cis,cis*-muconic acid, DAHP 3-deoxy-ᴅ-arabinoheptulosonate 7-phosphate,
DHQ 3-dehydroquinate, DHS 3-dehydroshikimate, E4P erythrose 4-phosphate,
FMN flavin mononucleotide, PCA protocatechuic acid, PEP phosphoenolpyruvate,
prFMN prenylated flavin mononucleotide. Source data are provided as a Source
Data file.

respectively ($\mu_{max}$ 0.12 h$^{-1}$ and OD$_{max}$ 1.3) relative to wild-type *P. occidentalis*. While these findings are indicative of CCM toxicity, our CCM
production strain is more extensively genetically modified than our
PCA-producing strain, raising the possibility that growth inhibition
could result from effects of genome editing, such as vector integration
or metabolic burden from increased expression of *cas9*.

To better establish the effect of CCM on *P. occidentalis* fitness, we
compared growth curves of wild-type *P. occidentalis* in 3× YNB saturated with CCM (<2 g L$^{-1}$) or adipic acid (22 g L$^{-1}$). In both instances the
pH of the medium was <3.5. While growth of *P. occidentalis* was largely
unaffected in the presence of 22 g L$^{-1}$ adipic acid ($\mu_{max}$ 0.28 h$^{-1}$ and
OD$_{max}$ 1.8) relative to media without acid supplementation ($\mu_{max}$
0.32 h$^{-1}$ and OD$_{max}$ 1.9), supplementation with <2 g L$^{-1}$ CCM caused a
38% and 42% reduction in maximal growth rate and OD$_{600}$ ($\mu_{max}$

0.20 h$^{-1}$ and OD$_{max}$ 1.1) (Supplementary Fig. 11), demonstrating significant toxicity of CCM relative to adipic acid. CCM toxicity was not
specific to *P. occidentalis* Y-7552, as all adipic acid tolerant *Pichia*
species and strains assayed exhibited marked toxicity to <2 g L$^{-1}$ CCM
(Supplementary Fig. 12) despite notable tolerance to adipic acid
(Fig. 3a, Table 1). Growth of *P. fermentans* 562 NCYC and the *S. cerevisiae* control was completely inhibited in the presence of <2 g L$^{-1}$ CCM.

## Discussion

In this study we screened 108 yeast strains across 28 genera and
identified *P. occidentalis* as a non-conventional host with exceptional
tolerance to several dicarboxylic acids, notably adipic acid. We
developed a comprehensive strain engineering toolkit for domestication of *P. occidentalis*, including techniques for CRISPR-Cas9 genome
editing, antibiotic marker recycling and high-level expression of heterologous genes. We leveraged our toolkit to engineer the host for
high-level production of CCM, the direct precursor to adipic acid,
using a simple mineral medium in a controlled fed-batch cultivation.
Under these conditions, production of CCM by our strain (38.8 g L$^{-1}$ at
0.511 g L$^{-1}$ h$^{-1}$) outperformed all prior *S. cerevisiae* CCM strains spanning more than 10 studies and achieving up to 22.5 g L$^{-1}$ CCM at
0.21 g L$^{-1}$ h$^{-1}$ (ref. 10). Our strain also outcompeted a recent evolved
and engineered *Pseudomonas putida* host with a CCM output of
33.7 g L$^{-1}$ CCM at 0.18 g L$^{-1}$ h$^{-1}$ (ref. 8). Commercial-scale production of
biobased acids typically demands titers of at least 50–100 g L$^{-1}$ at a rate
of more than 1 g L$^{-1}$ h$^{-1}$ (refs. 33,34) and, accordingly, our efforts provide a promising foundation for industrial production of CCM and
adipic acid using non-conventional yeasts, such as *P. occidentalis*.

CCM is converted to adipic acid via hydrogenation by bacterial
enoate reductase enzymes[35], which presently function poorly in
yeast[13]. Accordingly, *P. occidentalis* was selected for its robust tolerance to adipic acid, yet we engineered the strain to produce CCM due
to the lack of a suitable enoate reductase. Unexpectedly, CCM was
substantially more toxic to *P. occidentalis* than adipic acid, even when
adipic acid was supplied at a 10-fold higher concentration (22 g L$^{-1}$
compared to <2 g L$^{-1}$). CCM toxicity was pH-dependent, as growth of
strain LP635 could not be sustained between pH 2.0 and 3.5, well below
the pK$_a$ of CCM (3.87), yet was largely unaffected at pH 6.0 (OD$_{600}$ of
327 in fed-batch fermentation), enabling highly productive synthesis of
CCM (38.8 g L$^{-1}$). A previous fed-batch study employing engineered *S. cerevisiae* also reported growth arrest and incomplete sugar utilization
at CCM concentrations above 2.5 g L$^{-1}$ (ref. 6). In light of these findings,
we screened our most adipic acid tolerant yeasts for CCM tolerance,
which revealed that all three *P. occidentalis* isolates were the most
tolerant strains to both acids, suggesting similar mechanisms of toxicity between CCM and adipic acid. Owing to the unexpected toxicity of
CCM against all yeasts assayed, extending the heterologous pathway
to adipic acid is likely to improve strain fitness and metabolite production at low pH. Therefore, future efforts should focus on the
identification or engineering of alternative enoate reductase variants
with improved functionality in yeast hosts. Meanwhile, in situ extraction is a promising avenue to mitigate CCM toxicity, as a wide array of
biocompatible extractants have been evaluated for efficient removal of
CCM and related metabolites[6,36].

Other groups have shown that certain strains of *Pichia* are tolerant
to high concentrations of organic acids[37,38], yet the numbers of strains
assayed were comparatively small and comprised largely of environmental isolates rather than common type strains. Aside from one study
investigating lactic acid production using a novel isolate[37], *P. occidentalis* has not been widely utilized as a metabolic engineering host
and investigations into its acid tolerance have focussed largely on issues
with food contamination[39,40]. It is thus unlikely that *P. occidentalis* would
have been selected for screening had we reduced the breadth of our
assay or selected strains based on a simple literature search. Moreover,
our screen identified three strains of *P. occidentalis* as the most tolerant

to adipic acid of all 108 strains screened. Using *P. occidentalis* Y-7552, we developed methods for transformation and CRISPR-mediated genome editing. Owing to the low transformation efficiency and homologous recombination of *Pichia* spp., we designed a CRISPR-Cas9 delivery system in which homology-directed repair donors are pre-cloned into pCas vectors. Relatedly, the efficiency of CRISPR-mediated genome editing in *Pichia pastoris* was improved up to 25-fold by including an ARS within DNA repair donors[41]. Using this system, repair donors are stably inherited by daughter cells and are resistant to DNA exonucleases, thus increasing the likelihood of targeted editing events[42]. While we succeeded in inactivating the *ADE2* and *FCY1* loci, yielding mutants with screenable phenotypes (red pigmentation and 5-FC resistance, respectively), we were unable to identify a replicative plasmid origin for use in *P. occidentalis* despite the screening of several plasmids, including the broad *pan*ARS origin[43]. To address this challenge, we developed a CRISPR-mediated marker swapping strategy, which we leveraged to perform five successive rounds of genome editing in the construction of our final CCM production strain. To add to our genetic toolkit for *P. occidentalis*, we characterized a panel of 12 promoters and 12 terminators, which enabled us to modulate enzyme activity by a factor of 20. A genome-wide assay of all 5,302 *S. cerevisiae* terminators revealed an 81-fold difference in terminator activity[30], while the relative strength of *P. occidentalis* promoters ($P_{TPII} < P_{PGK1} < P_{TEF1} < P_{TDH3}$) followed a similar overall pattern as those from *P. pastoris* ($P_{PGK1} < P_{TPII} < P_{TEF1} < P_{TDH3}$)[44]. Taken together, our CRISPR-Cas9 editing technique and gene expression toolkit enables efficient strain engineering of *P. occidentalis* and contributes to the growing collection of non-conventional yeasts, such as *Yarrowia lipolytica*, *Kluyveromyces marxianus*, and *Komagataella phaffii* (*P. pastoris*)[45], that have been domesticated for industrial applications.

In summary, screening culture collections for prospective adipic acid tolerant yeasts identified *P. occidentalis* as a non-conventional and untapped host for industrial production of CCM and adipic acid. The strain engineering tools and techniques developed herein provide a solid foundation for engineering acid tolerant *Pichia* and will enable the construction of highly robust organic acid production strains.

## Methods

### Plasmids, strains, and growth media

All 153 non-*Saccharomyces* yeast strains ordered in this study are listed in Supplementary Data 1. *Saccharomyces cerevisiae* CEN.PK2-7D (*MATα*) was utilized for assembling plasmids in vivo via gap repair. Unless stated otherwise all yeast cultures were grown in YPD (10 g L$^{-1}$ Bacto Yeast Extract, 20 g L$^{-1}$ Bacto peptone, 20 g L$^{-1}$ glucose). Recombinant *S. cerevisiae* strains were selected on YPD agar containing 400 μg mL$^{-1}$ G418 or 200 μg mL$^{-1}$ hygromycin B. Recombinant *Pichia* strains were selected on YPD agar containing 1200 μg mL$^{-1}$ G418, 600 μg mL$^{-1}$ hygromycin B, or 100 μg mL$^{-1}$ nourseothricin. Plasmids and engineered strains employed in this work are provided in Tables 2 and 3, respectively. Oligonucleotides utilized in this work are provided in Supplementary Data 2.

### Microtiter plate assay for tolerance to various acids

Strains were received from repositories and used to inoculate liquid yeast malt (YM) medium, containing 5 g L$^{-1}$ of peptone, 3 g L$^{-1}$ of yeast extract, 3 g L$^{-1}$ of malt extract and 10 g L$^{-1}$ of dextrose. Strains were grown at 30 °C until fully saturated (up to seven days) and subsequently stored in 15% glycerol at −80 °C.

For high-throughput acid tolerance assays, frozen strains were revived in batches of 30 in YPD medium at 30 °C. Once all cultures were visibly turbid, strains were back-diluted (1:20) into 180 μL of YPD in a shallow 96-well plate and left to grow for 8 h or until cultures reached an OD$_{600}$ of 1.8. Following the second incubation, OD$_{600}$ was measured on a plate reader and a custom script was used to create a CSV file with the absolute volumes of diluent and cell culture necessary to dilute each culture to an OD$_{600}$ of 1.8. Dilutions were then carried out using the Span-8 of a Biomek FXP liquid handling system to create an intermediate plate with 30 strains in triplicate, including *S. cerevisiae* CEN.PK 113-7D as a control. From the intermediate plates, a 10 μL inoculum was drawn and dispensed into an experimental plate containing 170 μL of the relevant growth medium. The final volume of each well was 180 μL and the initial OD$_{600}$ of all strains was 0.1. Initial tests for organic acid tolerance were conducted in YPD medium, YPD$_{cit}$

## Table 2 | Plasmids utilized in this study

| Plasmid | Description | Source or reference |
|---|---|---|
| pCAS-G418-2μ | $P_{RNR2}$-*cas9$_{NLS}$*-T$_{CYC1}$, pUC, 2μ, $P_{tRNA\_Tyr}$–3'HDV-gRNA-Scaffold-T$_{SNR52}$, $P_{TEF1}$-*kanMX*-T$_{TEF1}$ | 46 |
| pCAS-Hyg-2μ | $P_{RNR2}$-*cas9$_{NLS}$*-T$_{CYC1}$, pUC, 2μ, $P_{tRNA\_Tyr}$–3'HDV-gRNA-Scaffold-T$_{SNR52}$, $P_{TEF1}$-*HphNTI*-T$_{TEF1}$ | 49 |
| pCas-G418-CEN6ARS4 | CEN6ARS4 derivative of pCAS-G418-2μ | This study |
| pCas-Hyg-CEN6ARS4 | CEN6ARS4 derivative of pCAS-Hyg-2μ | This study |
| pCas-G418-panARS | panARS derivative of pCAS-G418-2μ | This study |
| pCas-G418-CEN6ARS4-PoADE2 | *P. occidentalis ade2Δ* donor cloned into pCas-G418-CEN6ARS4 | This study |
| pCas-Hyg-CEN6ARS4-PoADE2 | *P. occidentalis ade2Δ* donor cloned into pCas-Hyg-CEN6ARS4 | This study |
| pCas-G418-2μ-PoADE2 | *P. occidentalis ade2Δ* donor cloned into pCAS-G418-2μ | This study |
| pCas-Hyg-2μ-PoADE2 | *P. occidentalis ade2Δ* donor cloned into pCas-Hyg-2μ | This study |
| pCas-G418-panARS-PoADE2 | *P. occidentalis ade2Δ* donor cloned into pCas-G418-panARS | This study |
| pCas-G418-NoOri-PoADE2 | Deletion of the 2μ origin from pCAS-G418-2μ-PoADE2 | This study |
| pCas-Hyg-CEN6ARS4-PklADE2 | *P. kluyveri ade2Δ* donor cloned into pCas-Hyg-CEN6ARS4 | This study |

## Table 3 | Engineered *Pichia occidentalis* strains utilized in this study

| ID | Description | Parent | Manipulation | Reference |
|---|---|---|---|---|
| LP618 | Deletion of *ADE2* | WT *P. occidentalis* | *ade2Δ* | This study |
| LP620 | Integration of *PaAROZ* | WT *P. occidentalis* | *ARO4::PaAROZ* | This study |
| LP622 | Integration of *PoARO4$^{K225L}$* | LP620 | *01b-PoARO4$^{K225L}$–01b* | This study |
| LP630 | Integration of *Ec.aroB* and *Ec.aroD* | LP622 | *ARO3::Ec.aroB-Ec.aroD* | This study |
| LP632 | Integration of *Kp.aroY* and *ScPAD1* | LP630 | *03b-Kp.aroY-ScPAD1–03b* | This study |
| LP635 | Integration of *CaHQD2* and *Kp.aroY* | LP632 | *03a-CaHQD2-Kp.aroY–03a* | This study |

(YPD containing 0.1 M citric acid added to decrease the pH to 3.0), or YPD$_{AA}$ (YPD containing 20 g L$^{-1}$ of adipic acid, resulting in a pH of 3.7). Organic acid tests conducted in minimal media used a common base of 20 g L$^{-1}$ of glucose, 5.1 g L$^{-1}$ of YNB without amino acids and without ammonium sulfate, 5.0 g L$^{-1}$ of ammonium sulfate, and 0.15 M of relevant organic acid. HCl was added dropwise to a final pH of 2.8. After inoculation, plates were sealed with parafilm and incubated at 30 °C in Tecan Sunrise plate readers for 48 h. Plates were orbitally shaken for 15 min, allowed to rest for 5 min prior to OD$_{600}$ readings, and shaking was restarted.

For CCM tolerance assays, growth curves were generated in triplicate in 96-well microtiter plates containing 180 µl of medium. Strains engineered for PCA or CCM production were assayed in 3× YNB medium containing 2% glucose. For tolerance to exogenous CCM, CCM was added to 3× YNB at 2 g L$^{-1}$, heated gently at 50 °C for 4-6 hours and the resulting partially dissolved slurry was filtered. For tolerance to exogenous adipic acid, adipic acid was dissolved at 22 g L$^{-1}$ (0.15 M) with gentle heating. Cultures were inoculated using saturated overnight cultures to an initial OD$_{600}$ of roughly 0.1 and wrapped in Parafilm to minimize evaporation. Absorbance was measured at 600 nm every 20 min with a Sunrise absorbance microplate reader (Tecan) over the course of 30 h. Maximum specific growth rates ($\mu_{max}$ h$^{-1}$) were determined from triplicate cultures and were based on OD$_{600}$ readings.

## Yeast strain construction

Yeast genetic modifications were made via CRISPR-Cas9-mediated genomic integration[46,47] and in vivo DNA assembly[48]. Sequences of integration loci, gRNAs, ADE2, promoters, and terminators from adipic acid tolerant Pichia spp. were obtained from reported draft genome sequences[24]. Cas9 and gRNAs were delivered to Pichia strains using pCas-G418-CEN6ARS4 (ref. [46]) or pCas-Hyg-CEN6ARS4 (ref. [49]). For chromosomal integration in Pichia, homology repair donors were first cloned into pCas vectors via gap repair in S. cerevisiae. Homology repair donors were designed to possess roughly 800 bp of flanking homology to the Pichia genome, in addition to 40–50 bp of internal overlap and 40–50 bp of external overlap to BglII-digested pCas-G418-CEN6ARS4 for assembly in S. cerevisiae. pCas vectors harboring repair donor templates were recovered from S. cerevisiae, linearized with BsaI or NotI and 200 ng was used to transform Pichia strains with a five- to tenfold excess of a gRNA species targeting the Pichia genomic integration locus (600 ng) relative to a gRNA targeting chromosomal G418- or Hyg-resistance markers (120 ng) in a standard 50 µL lithium acetate transformation. gRNAs were retargeted using a two-step SOE PCR. Cells were heat-shocked at 42 °C for 40 min, recovered for 16 h without shaking, and plated onto YPD agar plates containing appropriate antibiotics. Selection of gRNAs targeting Pichia loci was performed using CCTop[50]. gRNA target sites utilized in this study are provided in Table 4. Large intergenic regions (>1.5 kb) were selected as loci for DNA integration in P. occidentalis. P. occidentalis promoters (0.6–1.5 kb) were identified by selecting the full sequence flanked by the target and upstream coding sequences. P. occidentalis terminators (0.3–1.2 kb) were identified by selecting the full sequence flanked by the target and downstream coding sequences. Pichia expression cassettes are provided in Table 5. Sequences of P. occidentalis promoters and terminators, and integration loci are provided in Supplementary Data 3 and 4, respectively. Genes and synthetic DNAs are provided in Supplementary Data 5.

## Microtiter plate assay of GFP-producing strains

Pichia colonies expressing GFP variants were picked in triplicate into standard V-bottom 96-well microtiter plates containing 0.15 mL YNB medium and 20 g L$^{-1}$ glucose. Cultures were grown overnight and diluted into 0.8 mL fresh YNB medium in deep 96-well plates to an initial OD$_{600}$ of approximately 0.1. Cultures were

incubated with shaking for 12 h prior to OD$_{600}$ and fluorescence measurements. Fluorescence was measured using the M200 plate reader (Tecan) using an excitation wavelength of 485 nm and an emission wavelength of 525 nm. Gain was adjusted manually for

### Table 4 | Pichia target sites utilized in this study

| Organism | Target site ID | Target site sequence$^a$ | Reference |
|---|---|---|---|
| P. kluyveri | ADE2 | TAGAGAAAATGTCTGATTAAT<u>TGG</u> | This study |
| P. occidentalis | ADE2 | TTAATGAACCCGGTTGTTGAT<u>GG</u> | This study |
| | FCY1 | TAAGTTTCTGTTGTCGCCCT<u>GGG</u> | This study |
| | ARO3 | ATCAAGGGCAAACTTCCTTG<u>TGG</u> | This study |
| | ARO4 | TGACATCTTGTGAATCTGTGT<u>GG</u> | This study |
| | 01b | TTTTGGTAGCGTGCCTTGTAC<u>GG</u> | This study |
| | 03a | TGACTATACACTCTGTCTACA<u>GG</u> | This study |
| | 03b | TTTGTGTAAAACTATTAAGCG<u>GG</u> | This study |
| Other | G418$^R$ | TCCGTACTCCTGATGATGCAT<u>GG</u> | This study |
| | Hyg$^R$ | TCTCGATGAGCTGATGCTTT<u>GGG</u> | This study |

$^a$PAMs are underlined.

### Table 5 | Expression cassettes utilized in this study

| Description | Cassette$^a$ | Locus |
|---|---|---|
| eGFP | P$_{TDH3}$-eGFP-T$_{RPL3}$ | FCY1 |
| SuperfolderGFP (sfGFP) | P$_{TDH3}$-sfGFP-T$_{RPL3}$ | FCY1 |
| Envy GFP | P$_{TDH3}$-Envy-T$_{RPL3}$ | FCY1 |
| mNeonGreen GFP | P$_{TDH3}$-mNeonGreen-T$_{RPL3}$ | FCY1 |
| P$_{ASC1}$-mNeonGreen | P$_{ASC1}$-mNeonGreen-T$_{RPL3}$ | FCY1 |
| P$_{CYC1}$-mNeonGreen | P$_{CYC1}$-mNeonGreen-T$_{RPL3}$ | FCY1 |
| P$_{FBA1}$-mNeonGreen | P$_{FBA1}$-mNeonGreen-T$_{RPL3}$ | FCY1 |
| P$_{GPM1}$-mNeonGreen | P$_{GPM1}$-mNeonGreen-T$_{RPL3}$ | FCY1 |
| P$_{HHF1}$-mNeonGreen | P$_{HHF1}$-mNeonGreen-T$_{RPL3}$ | FCY1 |
| P$_{HHF2}$-mNeonGreen | P$_{HHF2}$-mNeonGreen-T$_{RPL3}$ | FCY1 |
| P$_{HTB2}$-mNeonGreen | P$_{HTB2}$-mNeonGreen-T$_{RPL3}$ | FCY1 |
| P$_{PGK1}$-mNeonGreen | P$_{PGK1}$-mNeonGreen-T$_{RPL3}$ | FCY1 |
| P$_{RNR2}$-mNeonGreen | P$_{RNR2}$-mNeonGreen-T$_{RPL3}$ | FCY1 |
| P$_{TEF1}$-mNeonGreen | P$_{TEF1}$-mNeonGreen-T$_{RPL3}$ | FCY1 |
| P$_{TPI1}$-mNeonGreen | P$_{TPI1}$-mNeonGreen-T$_{RPL3}$ | FCY1 |
| mNeonGreen-T$_{ASC1}$ | P$_{GPM1}$-mNeonGreen-T$_{ASC1}$ | FCY1 |
| mNeonGreen-T$_{CYC1}$ | P$_{GPM1}$-mNeonGreen-T$_{CYC1}$ | FCY1 |
| mNeonGreen-T$_{EFM1}$ | P$_{GPM1}$-mNeonGreen-T$_{EFM1}$ | FCY1 |
| mNeonGreen-T$_{ENO2}$ | P$_{GPM1}$-mNeonGreen-T$_{ENO2}$ | FCY1 |
| mNeonGreen-T$_{HTB2}$ | P$_{GPM1}$-mNeonGreen-T$_{HTB2}$ | FCY1 |
| mNeonGreen-T$_{IDP1}$ | P$_{GPM1}$-mNeonGreen-T$_{IDP1}$ | FCY1 |
| mNeonGreen-T$_{PGK1}$ | P$_{GPM1}$-mNeonGreen-T$_{PGK1}$ | FCY1 |
| mNeonGreen-T$_{RNR2}$ | P$_{GPM1}$-mNeonGreen-T$_{RNR2}$ | FCY1 |
| mNeonGreen-T$_{SRP54}$ | P$_{GPM1}$-mNeonGreen-T$_{SRP54}$ | FCY1 |
| mNeonGreen-T$_{TPI1}$ | P$_{GPM1}$-mNeonGreen-T$_{TPI1}$ | FCY1 |
| mNeonGreen-T$_{VMA2}$ | P$_{GPM1}$-mNeonGreen-T$_{VMA2}$ | FCY1 |
| PaAROZ | P$_{TDH3}$-PaAROZ-T$_{RPL3}$ | ARO4 |
| PoARO4$^{K225L}$ | P$_{TDH3}$-PoARO4$^{K225L}$-T$_{EFM1}$ | 01b |
| Ec.aroB-Ec.aroD | P$_{TDH3}$-Ec.aroB-T$_{IDP1}$-LTP1-P$_{GPM1}$-Ec.aroD-T$_{VMA2}$ | ARO3 |
| Kp.aroY-ScPAD1 | P$_{TDH3}$-Kp.aroY-T$_{SRP54}$-LTP1-P$_{TEF1}$-ScPAD1-T$_{RPL15A}$ | 03b |
| CaHQD2-Kp.aroY | P$_{TDH3}$-CaHQD2-T$_{SRP54}$-LTP1-P$_{TEF1}$-Kp.aroY-T$_{RPL15A}$ | 03a |

$^a$LTP1 is a synthetic 40 bp linker (5'-CCGGTCTTAGAAAACGCATAAACATACAAGTGGACAGATG-3') used to join expression cassettes.

each sample. A strain lacking GFP was included as a control for autofluorescence.

## Microtiter plate assay

*Pichia* colonies synthesizing heterologous metabolites were picked in triplicate into deep 96-well plates containing 0.5 mL YPD medium. Cultures were grown overnight, diluted 50× into fresh YPD medium, and incubated for 48-72 h. $OD_{600}$ of cultures was measured using the M200 plate reader (Tecan) and samples were frozen at −20 °C prior to HPLC or LC-MS analysis.

## Fed-batch cultivation of *P. occidentalis* strains

Controlled fed-batch fermentations were carried out in 3 L BioBundle fermentors (Applikon). Cultivation temperature was maintained at 30 °C. Fermentations were performed at pH 6.0 by automated titration with 12.5% $NH_3$ or without pH control and stepwise adjustment of pH during cultivation with 4 N NaOH to rescue growth of CCM-producing strains at low pH (pH 2.0-3.5). Dissolved oxygen was maintained at 30% of air saturation (aeration rate 1.5 L $min^{-1}$). Off-gas composition (concentration of $O_2$ and $CO_2$) was analyzed using a Tandem Multiplex gas analyzer (Magellan BioTech). Bioreactor inoculum was generated in a 500 mL shake flask containing 50 mL of batch medium (40 g glucose, 2.5 g $KH_2PO_4$, 6.0 g $(NH_4)_2SO_4$, 1.0 g $MgSO_4·7H_2O$, 5 mL vitamin stock, and 5 mL trace element stock per liter) and grown for 36-48 h at 30 °C. Composition of the vitamin and trace element stock solutions are described in Pyne et al. [51]. Cells were washed and suspended in 0.9% NaCl and used to inoculate ($OD_{600}$ = ~0.3) 1 L of batch medium. The culture was operated in batch mode until glucose was exhausted, followed by fed-batch phase with automated addition of feed. Feeding medium for cultivation at pH 6.0 contained 600 g glucose, 20.8 g $KH_2PO_4$, 5 g $(NH_4)_2SO_4$, 8.3 g $MgSO_4·7H_2O$, 5 g $K_2SO_4$, 20 mL vitamin stock, and 20 mL trace element stock per liter. Feeding medium for cultivation without pH control contained 360 g glucose, 7.5 g $KH_2PO_4$, 30 g $(NH_4)_2SO_4$, 3 g $MgSO_4·7H_2O$, 7.5 mL vitamin stock, and 7.5 mL trace element stock per liter. Samples were collected every 12 h for a total of three days. Cell dry weight (g $L^{-1}$) was calculated using a conversion factor of 0.36 g $L^{-1}$ per $OD_{600}$ (determined gravimetrically). Bioreactor data were collected and analyzed using BioXpert 1.3 (Applikon).

## LC-MS and HPLC-UV analysis

LC-MS was utilized to quantify PCA and CCM from cultures grown in deep 96-well plates. To extract metabolites, 7 μL cell broth was combined with 27 μL 100% acetonitrile and samples were shaken vigorously for 5 min, followed by addition of 146 μL of 0.123 % formic acid (0.1% final concentration). Extracts were centrifuged at 4000 RCF for 10 min and supernatants were further diluted 10-fold in 15% acetonitrile containing 0.1% formic acid prior to LC-MS analysis. Ten μL of diluted extract was separated on a 1290 Infinity II LC system (Agilent Technologies) with a Zorbax Rapid Resolution HT C18 column (50 × 2.1 mm, 1.8 μm; Agilent Technologies). Metabolites were separated using the following gradient: 2% B to 10% B from 0 to 4 min (0.3 mL $min^{-1}$), 10% B to 85% B from 4 to 6 min (0.3 mL $min^{-1}$), held at 85% B from 6 to 7 min (0.3 mL $min^{-1}$), 85% B to 2% B from 7 to 7.1 min (0.3 mL $min^{-1}$), and held at 2% B from 7.1 to 9 min (0.45 mL $min^{-1}$). Solvent A was 0.1% formic acid in water and solvent B was 0.1% formic acid in 100% ACN. Following separation, eluent was injected into an Agilent 6545 quadrupole time-of-flight MS (QTOF-MS; Agilent Technologies) in negative mode. The sample tray and column compartment were set to 4 °C and 30 °C, respectively. The sheath gas flow rate and temperature were adjusted to 10 L $min^{-1}$ and 350 °C, respectively, while drying and nebulizing gases were set to 12 L $min^{-1}$ and 55 psig, respectively. The drying gas temperature was set to 325 °C. QTOF data was processed and manipulated using MassHunter Workstation software version B.06.00 (Agilent Technologies).

Glucose, glycerol, PCA, catechol, and CCM from fed-batch fermentation broth samples were analyzed and quantified using HPLC-UV and RID. For analysis of CCM, culture broth was combined with an equal volume of 10 N NaOH and incubated for 10 min with shaking. Cell debris was centrifuged at maximum speed for 10 min and supernatant was diluted in 40 mM PBS, pH 7.4. Alternatively, CCM was analyzed by diluting culture supernatant 50-fold directly in water or 40 mM PBS, pH 7.4. For analysis of glucose, glycerol, PCA, and catechol, culture broth was diluted in 40 mM PBS, pH 7.4 and centrifuged at maximum speed for 10 minutes. Five μL of extracted broth was separated on an Agilent 1200 HPLC system equipped with an Aminex Fast Acid Analysis column. Metabolites were separated isocratically using 10 mM $H_2SO_4$ at a flow rate of 0.8 mL $min^{-1}$. PCA and catechol were detected at 210 nm. CCM was detected at 260 nm. Traces were collected and analyzed using ChemStation version B.02.00 (Agilent Technologies).

## Statistical analyses

All numerical values are depicted as means ± s.d. Statistical differences between control and engineered strains were assessed via two-tailed Student's *t*-tests assuming equal variances. In all cases, *P*-values < 0.05 were considered significant. Statistical significance of dicarboxylic acid chain length was assessed via OLS linear regression with hourly growth rates as the response variable. Statistical analyses were performed using Excel 2306 Build 16.0.16529.20164 (Microsoft) or Prism version 9.5.1 (GraphPad).

## Reporting summary

Further information on research design is available in the Nature Portfolio Reporting Summary linked to this article.

## Data availability

Draft genome sequences of relevant acid tolerant *Pichia* strains were accessed under the BioProject accession PRJNA870168. Source data are provided with this paper.

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

## Acknowledgements

We thank Nicholas Gold and the Concordia Genome Foundry for assistance with high-throughput screening. We also thank Marcos DiFalco for assistance with LC-MS analyses. This study was financially supported by an NSERC-Industrial Biocatalysis Network (IBN) grant, an NSERC Discovery grant, and by bp Biosciences Center. M.E.P. was supported by an NSERC Postdoctoral Fellowship, K.K. was supported by a Concordia University Horizon Postdoctoral Fellowship, and L.N. was supported by a FQRNT DE Doctoral Research Scholarship for Foreign Students. V.J.J.M. is supported by a Concordia University Research Chair. M.W. is supported by a Canada Research Chair.

## Author contributions

M.E.P., J.A.B., L.N., K.K., Q.W., and V.J.J.M. designed the research. M.E.P., J.A.B., and L.N. performed the experiments. K.E. and M.D. assisted in preliminary studies and fed-batch cultivation. V.J.J.M. and Q.W. supervised the research. M.E.P., J.A.B., M.W. and V.J.J.M. wrote the manuscript.

## Competing interests

The authors declare no competing interests.
