## [Peer Review File · Nature Communications]

Screening non-conventional yeasts for acid tolerance and engineering *Pichia occidentalis* for production of muconic acidReviewers' Comments:

Reviewer #1:

Remarks to the Author:

This manuscript describes screening of high acid (or lower pH) tolerance yeast. They screened 108 publicly accessible yeast strains for having tolerance to adipic acid. After selection of *Pichia occidentalis* as a candidate host for *cis,cis*-muconic acid producer, they developed muconic acid producing *Pichia* strain and efficient muconic acid production was demonstrated.

1. This study is muconic acid production in yeast. However, host selection is based on adipic acid resistance as an indicator. Simply thinking, the screening should be done using muconic acid resistance as an indicator. Also, if screening is done using adipic acid as an indicator, the main goal of this study should be to develop yeast producing adipic acid as a final product. Throughout the Introduction, Results, and Discussion, these seem to mislead the readers.
2. Selection by growth is a common method, and it is not clear that the selected yeasts in this study are necessarily suitable hosts for substance production. The reasons why these yeasts are highly resistant should be elucidated.
3. Selection by growth is a common method, and it is not clear that the selected yeasts in this study are necessarily suitable hosts for substance production. The reasons why these yeasts are highly resistant should be elucidated.
4. Supplementary Figs 10-12; These results show that strain selection should be carried out using muconic acid resistance.
5. Even if dicarboxylic acid tolerance and pH tolerance are high, if the tolerance is reduced by metabolic engineering, something to improve is needed. In this respect, no advantage can be found to general hosts.

Reviewer #2:

Remarks to the Author:

The work of Pyne et al. developed, for the first time (in the species), the production of the add-value molecule CCM in the species *P. occidentalis*, selected based on its higher resilience to stress induced by carboxylic acids. This study is in line with several others demonstrating the usefulness of Yeasts for such purposes, highlighting the importance of production hosts harbouring mechanisms that allow them to cope with the toxic effects of the acids that they are producing. From the academic perspective, the knowledge herein developed is not much novel only adding another species (*P. occidentalis*) to the already portfolio available of production Yeasts that besides also showing production capabilities, exhibit high acid tolerance (this is, for example, the case of *I. orientalis*, now *P. kudriavzevii*, or *Z. bailii*; both two species represented in the cohort of species also tested by the authors). However, the achievements in other species were not specifically focused on CCM production and in that sense the knowledge herein developed will surely be impactful. In general I think the work presented in this MS is very carefully conducted and the MS shows in a very organized manner the line of thinking that originated this research, along with the development of tools that may advance further metabolic engineering of this novel catalyst, *P. occidentalis*. I have relatively minor remarks to the results presented that I detail below:

- 1) Organic acids are very different in terms of their toxicity and many studies in *S. cerevisiae* have been showing this pointing to specific toxic effects that are caused by one acid and that are not caused by the others. The authors have themselves face a bit this issue when their production strain faces problems with toxicity caused by CCM. In this sense throughout the paper it would probably be

beneficial not to use "acid tolerance" but instead specificity to which tolerance is the figure/table/sentence referring to (e.g. in Fig.1 when the authors put a legend stating organic acid tolerance of the different yeasts, what was tested was Citrate and AA tolerance; and the same also in Fig.3).

2) A consequence of the different toxic effects of different organic acids is the high toxicity exerted by CCM over *Pichia* production cells, that tolerate extremely well AA. Since the 1st cohort of strains is selected based on their capability to withstand stress caused by adipic acid, compared to citric acid; we actually don't know if the behaviour of the overall cohort of ~180 strains tested would be identical had they been profiled first for their tolerance to CCM. In other words, by selecting for AA-tolerant species you may not necessarily be identifying the best-performing species/strain when it concerns tolerance to CCM, since these can exert quite different toxicity mechanisms. Do you have evidences pointing to similar toxicity mechanisms of these two acids that might led us to say that the selection you made for AA-tolerant strains will likely also result in tolerance to other acids that are also intermediates of the pathway?

2. In the second screening the authors use the same concentration of three organic acids and the same pH and afterwards they make a discussion on the toxic effects associated with the different structures. however, this is only reasonable if the undissociated amounts of each of these organic acids is the same (which necessarily will depend on the different pKas). Thus, in order to make these structure-phenotype associations the specific amount of the different undissociated acid forms should be presented and compared with for the sake of a more rigorous comparison. Again, a similar comparison is performed with monocarboxylic acids but this is only reasonable if we are comparing similar amounts of undissociated acid forms. In other words, the use of equimolar concentrations of the different organic acids does not assure that you have the same concentration of undissociated acid, which is the only toxic form.

3. Often the very high tolerance of Yeast to certain organic acids results from them having the capability of consuming the corresponding acid anion. Had the authors checked that in the AA tolerance assays the acid is still present in similar concentrations as they were added in the beginning? Is it possible that these Yeasts are equipped with adipate consumption pathways? This would be impactful for their use as hosts!

4. What is the capability of the genetic tools herein developed to be transferred to other strains of *P. occidentalis*? This issue comes from the relative phenotypic variation observed across several traits including resistance to antimicrobials used for selection and also "acid tolerance". Was the production phenotype herein established ported to to other *P. occidentalis* strains with success or is this something strain-specific? To boost *P. occidentalis* as novel microbial catalyst would require tools that can used across.

5. In a paper that addresses the issue of organic acid toxicity, the beginning of the discussion section seems a bit out of focus by showing surprise to the expected effects of pH over adipate/adipic acid toxicity in *E. coli* (more than expected...) and also the different aspects of organic acid toxicity (that have been described in many more studies, including with higher depth, than the single reference mentioned by the authors). I found surprising that the authors only approach this issue here, in the discussion section.

6. All the production pathway was obtained in a mineral medium containing vitamins and trace elements solution. This will be, necessarily, limitative to be utilized at an industrial scale so i was wondering whether the authors had attempted or consider to try production with other growth medium that would not require this exquisite (and expensive) set of nutrients! That would be most impactful!

Reviewer #3:

Remarks to the Author:

Pyne and co-authors present an excellent study in developing a *P. occidentalis* system for the production of muconic acid. They not only conduct a thoughtful survey of host systems with desirable traits for production of diacids (tolerance), but then also develop an important suite of tools in this microbe. The tools range from CRISPR-based editing, plasmid systems, promoters, and terminators as well as information on antibiotic markers that can be used. The efforts to establish a known muconic acid heterologous pathway were also done very well leading to impressive and the highest to-date titer rate and yield metrics for muconic acid from glucose. This is an impactful study and advances work in this host and product considerably,

My comments are as follows:

For muconic acid, what the titer rate and yield are actually needed for the bioconversion process to be viable for commercialization? The authors reach impressive values, but is this enough? It is important for readers to have that insight. Please conduct some rudimentary techno-economic analyses to provide this discussion.

Is production from glucose sufficient? The authors cite Ling et al 2022 where the production of muconic acid was shown in *P. putida* at very high titer rates and yields and from glucose and xylose. How does this present study compare to that? Would it be useful to show xylose conversion in *P. occidentalis* also? What other carbon sources can *P. occidentalis* use?

Minor comment: Please clearly state in the results that production assays were done in defined medium as this is important.

Response to Reviewer's Comments for NCOMMS-23-17832

Screening non-conventional yeasts for organic acid tolerance and engineering *Pichia occidentalis* for production of *cis,cis*-muconic acid

Reviewer #1 comments:

Comment 1: “This study is muconic acid production in yeast. However, host selection is based on adipic acid resistance as an indicator. Simply thinking, the screening should be done using muconic acid resistance as an indicator. Also, if screening is done using adipic acid as an indicator, the main goal of this study should be to develop yeast producing adipic acid as a final product. Throughout the Introduction, Results, and Discussion, these seem to mislead the readers.”

Response to comment 1: We thank the reviewer for this suggestion. It is correct that we screened for adipic acid (AA) tolerance, yet engineered muconic acid (CCM) production. However, it is important here to re-emphasize the big picture of our work, which is biobased production of AA, not CCM. Presently, there is no known enzyme capable of efficiently converting CCM to AA in yeast, as characterized enoate reductase (ER) variants from bacteria yield nearly undetectable amounts of AA (2.59 mg/L) in yeast¹. If we had screened for CCM tolerance alone, our strain would potentially not be the best AA host following the identification of a suitable ER enzyme, requiring that we repeat all our screening and engineering efforts using AA in the future. We are hopeful that an efficient ER enzyme will soon be discovered or engineered, which uniquely positions *P. occidentalis* for highly efficient production of AA at low pH.

In the absence of an active ER, our strain produces CCM, which is ultimately a temporary end product. Further, we did screen a subset of AA-tolerant *Pichia* for CCM tolerance, which is provided in Supplementary Fig. 12. Importantly, a similar trend of relative tolerance was observed between AA and CCM (*P. occidentalis* > *P. kluyveri* > *P. manshurica* > *P. fermentans* = *S. cerevisiae*), which suggests that CCM tolerance parallels AA tolerance. Ultimately, *P. occidentalis* was the most AA tolerant ($n = 108$) and CCM tolerant ($n = 7$) of strains screened. It is also clear that CCM is comparatively much more toxic than AA, even at 2 g/L vs. 20 g/L, but again CCM is a temporary end product pending discovery of a superior enoate reductase variant.

Changes made in text: We clarified potential confusion surrounding this issue in the Introduction (page 4), Results (page 18), and Discussion (page 23).

Comment 2: “Selection by growth is a common method, and it is not clear that the selected yeasts in this study are necessarily suitable hosts for substance production. The reasons why these yeasts are highly resistant should be elucidated.”

Response to comment 2: We thank the Reviewer for this suggestion and agree that the mechanism behind acid tolerance is interesting. However, elucidating tolerance mechanisms is beyond the intended scope of this paper, and we do not wish to speculate without having conducted thorough experiments. Instead, our objectives of the current contribution were to **1)** screen diverse non-*Saccharomyces* yeast for adipic acid tolerance, **2)** develop genome editing and strain engineering tools for acid tolerant strains, **3)** engineer a highly tolerant strain for CCM production, and **4)** characterize our best CCM-producer in a fed-batch process using a simple mineral medium. Many efforts have focused on elucidating acid tolerance mechanisms in yeast

and other microbes, and future efforts will likely focus on probing these mechanisms in *Pichia* and other non-conventional hosts.

Changes made in text: No changes made.

Comment 3: “Supplementary Figs 10-12; These results show that strain selection should be carried out using muconic acid resistance.”

Response to comment 3: Our response to Comment #1 above is relevant here. Again, we wish to emphasize the big picture, which is biobased production of AA, not CCM. Our work does highlight the somewhat unexpected toxicity of CCM, but screening for CCM tolerance alone would be short-sighted, as CCM is not the target end product, and we are hopeful that an efficient enoate reductase enzyme will be sourced in the near future. Our strain is clearly the most AA-tolerant strain of all 108 yeasts screened and is the ideal yeast host for biobased production of AA.

Changes made in text: See changes outlined in Comment #1 above. We also added max growth rates (μ_{\max}) and area under the curve (AUC) calculations for Supplementary Figure 12 to show that all three *P. occidentalis* strains were the most tolerant to CCM.

Comment 4: “Even if dicarboxylic acid tolerance and pH tolerance are high, if the tolerance is reduced by metabolic engineering, something to improve is needed. In this respect, no advantage can be found to general hosts.”

Response to comment 4: We presume that the Reviewer is referring to data in Supplementary Fig. 12a, which shows a reduction in growth/fitness for *P. occidentalis* strains engineered for PCA and CCM production. This is not a reduction in tolerance, as the reduction in growth likely results from the production of toxic metabolites, most notably CCM. Our *P. occidentalis* strain is still far more tolerant to CCM than *S. cerevisiae* and *P. fermentans* (see Supplementary Fig. 12). A decline in cell fitness is a common and even anticipated phenotype of highly engineered microbes (see Supplementary Fig. 3 of ref. ²), especially strains engineered to produce toxic products. These strains have also been subjected to multiple rounds of CRISPR-Cas9 genome editing, heterologous DNA integration, and selection with yeast antibiotics (G418 and hygromycin), which likely also result in a burden to the cell relative to the wild-type strain.

Changes made in text: Our changes outlined in previous comments #1 and #3 should also help to clarify this issue.

Reviewer #2 comments:

Comment 1: “Organic acids are very different in terms of their toxicity and many studies in *S. cerevisiae* have been showing this pointing to specific toxic effects that are caused by one acid and that are not caused by the others. The authors have themselves face a bit this issue when their production strain faces problems with toxicity caused by CCM. In this sense throughout the paper it would probably be beneficial not to use “acid tolerance” but instead specificity to which tolerance is the figure/table/sentence referring to (e.g. in Fig.1 when the authors put a legend stating organic acid tolerance of the different yeasts, what was tested was Citrate and AA tolerance; and the same also in Fig.3).”

Response to comment 1: We thank the reviewer for this suggestion.

Changes made in text: We specified which acid we were referring to each time we used the term tolerance throughout the manuscript (see Track Changes). Fig. 3 was modified, as well as the legends to Figures 1-3.

Comment 2: “A consequence of the different toxic effects of different organic acids is the high toxicity exerted by CCM over *Pichia* production cells, that tolerate extremely well AA. Since the 1st cohort of strains is selected based on their capability to withstand stress caused by adipic acid, compared to citric acid; we actually don't know if the behaviour of the overall cohort of ~180 strains tested would be identical had they been profiled first for their tolerance to CCM. In other words, by selecting for AA-tolerant species you may not necessarily be identifying the best-performing species/strain when it concerns tolerance to CCM, since these can exert quite different toxicity mechanisms. Do you have evidences pointing to similar toxicity mechanisms of these two acids that might led us to say that the selection you made for AA-tolerant strains will likely also result in tolerance to other acids that are also intermediates of the pathway?”

Response to comment 2: This is a very relevant and insightful question. Our responses to Comments #1 and #3 from Reviewer #1 are relevant here. Because our target end product is AA, not CCM, we only screened a subset of the most adipic acid tolerant strains on CCM ($n = 7$), which is provided in Supplementary Figure 12. As stated above, CCM tolerance of *Pichia* strains followed the same general trend as AA tolerance (*P. occidentalis* > *P. kluyveri* > *P. manshurica* > *P. fermentans*), which provides some indirect evidence of similar toxicity mechanisms. Importantly, all three *P. occidentalis* isolates outperformed other AA tolerant strains when challenged with CCM, which is promising given our selection of *P. occidentalis* for AA production.

Changes made in text: We added a sentence at the end of the Results section (page 22) to clarify the similarities in relative toxicity observed between CCM and AA, which might indicate a similarity in mechanism of toxicity between the two acids. To better highlight this finding, we also added max growth rates (μ_{\max}) and area under the curve (AUC) calculations for Supplementary Figure 12 to show that all three *P. occidentalis* strains were the most tolerant to CCM. Based on these data, we suspect that *P. occidentalis* is likely to be one of the most, if not the most, tolerant strains to CCM if we screened our entire collection on CCM. However, if an efficient enoate reductase enzyme is discovered or engineered in the future, this point becomes irrelevant, as *P. occidentalis* is unequivocally the best AA strain of all 108 screened in this study.

Comment 3: “In the second screening the authors use the same concentration of three organic acids and the same pH and afterwards they make a discussion on the toxic effects associated with the different structures. however, this is only reasonable if the undissociated amounts of each of these organic acids is the same (which necessarily will depend on the different pKas). Thus, in order to make these structure-phenotype associations the specific amount of the different undissociated acid forms should be presented and compared with for the sake of a more rigorous comparison. Again, a similar comparison is performed with monocarboxylic acids but this is only reasonable if we are comparing similar amounts of undissociated acid forms. In other words, the use of equimolar concentrations of the different organic acids does not assure that you have the same concentration of undissociated acid, which is the only toxic form.”

Response to comment 3: This is a valid point, and we apologize for our oversimplification. Because the pKa values of these dicarboxylic acids are quite similar (succinic = 4.3 and 5.6; glutaric = 4.34 and 5.22; and adipic = 4.41 and 5.41), which reflects their structural similarity,

and our experiment was performed well below both pKa values of all acids (pH = 2.8), we believe that our assumption is valid and provides a close approximation of the relative toxicity of these acids.

Changes made in text: We added two statements clarifying our assumption of equal rates of dissociation (page 9).

Comment 4: *“Often the very high tolerance of Yeast to certain organic acids results from them having the capability of consuming the corresponding acid anion. Had the authors checked that in the AA tolerance assays the acid is still present in similar concentrations as they were added in the beginning? Is it possible that these Yeasts are equipped with adipate consumption pathways? This would be impactful for their use as hosts!”*

Response to comment 4: This is another valid point, but we did not test for adipic acid consumption given the number of strains (108), acids (5), growth media (2), and replicates (3) screened in this study. However, it is promising that *P. occidentalis* does not consume or transform CCM since we were able to synthesize nearly 40 g/L. In addition, we did not observe biocatalytic degradation products of CCM (i.e. hexenedioic acid) in our cellular extracts. Organic acids from the consumption of the corresponding acid anion is likely more relevant to production of natural organic acids, such as acetic, succinic, lactic, and citric acids, rather than adipic acid, with only a few studies reporting on this dicarboxylic acid being metabolized by bacteria using an adapted beta-oxidation pathway³.

Changes made in text: No changes were made.

Comment 5: *“What is the capability of the genetic tools herein developed to be transferred to other strains of *P. occidentalis*? This issue comes from the relative phenotypic variation observed across several traits including resistance to antimicrobials used for selection and also “acid tolerance”. Was the production phenotype herein established ported to other *P. occidentalis* strains with success or is this something strain-specific? To boost *P. occidentalis* as novel microbial catalyst would require tools that can used across.”*

Response to comment 5: To clarify, the three *P. occidentalis* strains discussed in the manuscript were comparable with respect to adipic acid tolerance, and the three strains outcompeted all of the remaining 105 strains assayed (see Figure 2a and 3a). However, the *P. occidentalis* isolates differed in genetic tractability, specifically transformation and antibiotic susceptibility, which guided our secondary selection criteria.

It is noteworthy that our tools and methodologies are transferrable between *P. occidentalis* 02W (NRRL Y-7552) and 04R (NRRL Y-6545), as we were also able to construct a PCA-producing strain of 04R (NRRL Y-6545), which is not included in our manuscript. When engineered with the same PCA pathway, 02W (NRRL Y-7552) and 04R (NRRL Y-6545) produced nearly identical amounts of PCA (~2 g/L), but we did not construct catechol- or CCM-producing strains of 04R (NRRL Y-6545) due to the ease of engineering our preferred 02W (NRRL Y-7552) strain. Transformation and DNA integration are less efficient in 04R (NRRL Y-6545) and 04Q (NRRL YB-3389), and we had issues consistently attaining transformants and/or CRISPR integrants of 04Q (NRRL YB-3389). Transformation of 04Q (NRRL YB-3389) could likely be improved by optimizing the heat shock parameters (temperature + duration) or by employing common transformation-promoting additives, such as DTT, DMSO, or 2-mercaptoethanol, in which case our *P. occidentalis* tools would likely prove to be completely transferrable. Finally, the *P. occidentalis* promoters, terminators, and integration loci that we

characterized are nearly identical between the three isolates, which will also promote strain engineering of all three *P. occidentalis* strains.

Changes made in text: No changes were made.

Comment 6: *“In a paper that addresses the issue of organic acid toxicity, the beginning of the discussion section seems a bit out of focus by showing surprise to the expected effects of pH over adipate/adipic acid toxicity in E. coli (more than expected...) and also the different aspects of organic acid toxicity (that have been described in many more studies, including with higher depth, than the single reference mentioned by the authors). I found surprising that the authors only approach this issue here, in the discussion section.”*

Changes made in text: Owing to these concerns and the current length of the manuscript, we reframed and removed portions of this paragraph (page 23).

Comment 7: *“All the production pathway was obtained in a mineral medium containing vitamins and trace elements solution. This will be, necessarily, limitative to be utilized at an industrial scale so i was wondering whether the authors had attempted or consider to try production with other growth medium that would not require this exquisite (and expensive) set of nutrients! That would be most impactful!”*

Response to comment 7: Our production medium was a simple, standard mineral medium containing chemically defined components, including small amounts of trace elements and vitamins, which is common practice for benchtop bioreactor trials. Comparable studies often supplement rich or complex media components to fed-batch fermentations, such as yeast extract or commercial yeast dropout supplements containing all 20 amino acids, to boost growth and production metrics. Our batch and fed-batch media were devoid of all complex undefined media components, and we did not supplement any amino acids to our fed-batch media. Interestingly, *P. occidentalis* can grow in vitamin-free medium⁴, yet we have not investigated our CCM production metrics in the absence of vitamins, although this would be a useful and illuminating next step.

Changes made in text: We added a new Table (**Supplementary Table 1**), which highlights the ability of *P. occidentalis* and *P. kudriavzevii* to grow in vitamin-free medium.

Reviewer #3 comments:

Comment 1: *“For muconic acid, what the titer rate and yield are actually needed for the bioconversion process to be viable for commercialization? The authors reach impressive values, but is this enough? It is important for readers to have that insight. Please conduct some rudimentary techno-economic analyses to provide this discussion.”*

Response to comment 1: We thank the reviewer for this suggestion. For biobased production of acids, it is widely held that titers > 50-100 g/L and productivities > 1 g/L/h are required^{5,6}.

Changes made in text: We added a statement in the Discussion section detailing these target metrics (page 23) and added the appropriate citations.

Comment 2: *“Is production from glucose sufficient? The authors cite Ling et al 2022 where the production of muconic acid was shown in P. putida at very high titer rates and yields and from*

glucose and xylose. How does this present study compare to that? Would it be useful to show xylose conversion in *P. occidentalis* also? What other carbon sources can *P. occidentalis* use?"

Response to comment 2: We thank the reviewer for this helpful suggestion. To clarify the ability of acid tolerant *Pichia* to utilize xylose, we performed an experiment by patching colonies of our top 8 tolerant *Pichia* strains onto agar plates containing xylose as sole carbon source. Of the 8 strains, only *P. fermentans* was able to grow on xylose as sole carbon source. Other substrate utilization data for thousands of yeast species is widely available online (theyeasts.org) and the corresponding publication³. We searched these resources for relevant data on our *Pichia* species, which supported our xylose utilization data, as *P. fermentans* is confirmed to utilize D-xylose. *P. occidentalis* is able to utilize glucose, glycerol, and ethanol, but is unable to grow on sucrose, galactose, lactose, xylose, starch, and cellobiose. Interestingly, *P. occidentalis* and *P. kudriavzevii* are relatively unique amongst *Pichia* in their ability to grow in vitamin-free medium.

Changes made in text: We added a column for our xylose utilization data to our 8 acid tolerant *Pichia* summary in Fig. 3a and modified the legend accordingly. We also added a few sentences describing substrate utilization and growth characteristics of *P. occidentalis* (page 13), as well as a new Table (**Supplementary Table 1**), which summarizes previous growth characteristics of all relevant acid tolerant *Pichia* species.

Comment 3: "Please clearly state in the results that production assays were done in defined medium as this is important."

Changes made in text: Clarified on pages 8+9.

Additional changes:

Alignment of axes in Figure 2b was modified.

References

- 1 Raj, K. *et al.* Biocatalytic production of adipic acid from glucose using engineered *Saccharomyces cerevisiae*. *Metabolic Engineering Communications* **6**, 28-32 (2018).
- 2 Pyne, M. E. *et al.* A yeast platform for high-level synthesis of tetrahydroisoquinoline alkaloids. *Nature Communications* **11**, 1-10 (2020).
- 3 Parke, D., M. A. Garcia & L. N. Ornston. Cloning and genetic characterization of *dca* genes required for β -oxidation of straight-chain dicarboxylic acids in *Acinetobacter* sp. strain ADP1. *Applied and Environmental Microbiology* **67**, 4817-4827 (2001).
- 4 Kurtzman, C., Fell, J. W. & Boekhout, T. *The yeasts: A taxonomic study*. (Elsevier, 2011).
- 5 Wang, J., Lin, M., Xu, M. & Yang, S.-T. Anaerobic fermentation for production of carboxylic acids as bulk chemicals from renewable biomass. *Anaerobes in Biotechnology*, 323-361 (2016).
- 6 Warnecke, T. & Gill, R. T. Organic acid toxicity, tolerance, and production in *Escherichia coli* biorefining applications. *Microbial Cell Factories* **4**, 1-8 (2005).

Reviewers' Comments:

Reviewer #1:

Remarks to the Author:

The authors adequately addressed this reviewer's concerns.

Reviewer #2:

Remarks to the Author:

I thank the authors for having modified the text of the MS in some points that, according with my opinion, were not so clear. I have no further comments.

Reviewer #3:

Remarks to the Author:

The authors have sufficiently addressed my comments.